# ECOMAN: an open-source package for geodynamic and seismological modelling of mechanical anisotropy.

Manuele Faccenda[1*], Brandon P. VanderBeek[1*], Albert de Montserrat[2], Jianfeng Yang[3], Francesco Rappisi[4], Neil Ribe[5]

[1]Dipartimento di Geoscienze, Università di Padova, 35131, Padova, Italy.
[2]Institute of Geophysics, ETH Zürich, Sonneggstrasse 5, 8092, Zürich, Switzerland.
[3]State Key Laboratory of Lithospheric Evolution, Institute of Geology and Geophysics, Chinese Academy of Sciences, Beijing, China.
[4]School of Earth and Environment, University of Leeds.
[5]Lab FAST, Univ Paris-Saclay, CNRS, Bat 530, rue André Rivière, F-91405 Orsay, France.

*Correspondence to*: Manuele Faccenda (manuele.faccenda@unipd.it), Brandon P. Vanderbeek (brandonpaul.vanderbeek@unipd.it)

**Abstract.** Mechanical anisotropy related to rock fabrics is a proxy for constraining the Earth's deformation patterns. However, the forward and inverse modelling of mechanical anisotropy in 3D large-scale domains has been traditionally hampered by the intensive computational cost and the lack of a dedicated, open-source computational framework. Here we introduce ECOMAN, a software package for modelling strain-/stress-induced rock fabrics and testing the effects of the resulting elastic and viscous anisotropy on seismic imaging and mantle convection patterns.

Differently from existing analogous software, ECOMAN can model strain-induced fabrics across all mantle levels and is optimised to run efficiently on multiple CPUs. It also enables modelling of shape preferred orientation (SPO)-related structures that can be superimposed over lattice/crystallographic preferred orientation (LPO/CPO) fabrics, which allows the consideration of the mechanical effects of fluid-filled cracks, foliated/lineated grain-scale fabrics and rock-scale layering.

One of the most important innovations is the Platform for Seismic Imaging (PSI), a set of programs for performing forward and inverse seismic modelling in isotropic/anisotropic media using real or synthetic seismic datasets. The anisotropic inversion strategy is capable of recovering parameters describing a tilted transversely isotropic (TTI) medium, which is required to reconstruct 3D structures and mantle strain patterns and to validate geodynamic models.

## 1 Introduction

The study of the Earth's interior has been traditionally based on seismological and geodynamic modelling, the former providing important information about its present-day structure, composition and state (Chang et al., 2015; French and Romanowicz, 2015; Schaeffer et al., 2016; Debayle et al., 2020), and the latter about its dynamics and compositional evolution (Davies et

al., 2012; Crameri and Tackley, 2014; Müller et al., 2022). Seismological and geodynamical modelling are very often conducted independently, which creates mechanical and geometrical inconsistencies across the models, hampers the interpretation of seismic observations in terms of geodynamic processes, and exacerbates the non-uniqueness of geodynamic model predictions. An alternative approach is combining computational seismology and geodynamics with mineral physics, which provides a comprehensive understanding of the Earth's interior processes, seismic behaviour, and material properties. This multidisciplinary methodology has been used in previous studies to post-process geodynamic flow calculations with thermodynamically self-consistent models of mantle mineralogy and converting thermal structure into isotropic elastic parameters. The obtained seismic mantle structure can then be used in simulations of global wave propagation, such that specific hypotheses on mantle dynamics can be tested directly against seismic data (Styles et al., 2011; Schubert et al., 2012; Maguire et al., 2018). The inverse procedure consists of converting seismic anomalies into density anomalies driving mantle flow models and is typically employed to quantify dynamic topography and mantle viscosity structure, and to reproduce large-scale mantle flow patterns (e.g., Bunge et al., 2003; Steinberger and Calderwood, 2006; Rudolph et al., 2015). However, isotropic seismic imaging provides limited information regarding local-/regional-scale dynamical processes (Fraters and Billen, 2021), and a better way to couple the geodynamic evolution and seismological structure of the Earth's interior is by accounting for the strain-/stress-induced mechanical anisotropy of crustal and mantle rocks.

Mechanical anisotropy refers to the directional dependence of mechanical properties in a material and is well known to affect both elastic and viscous deformational behaviour. Mechanical anisotropy depends on several factors, including the lattice or crystal preferred orientation (LPO, CPO) of intrinsically anisotropic minerals, and extrinsic mechanisms related to the shape preferential orientation (SPO) of melt, fluid, or air-filled fractures and non-spherical pores, and grain- or rock-scale compositional layering. Most micro- and macro-scale fabrics are acquired as a function of the cumulative deformation and material mechanical properties, and as such they constitute an important source of information about the Earth's dynamical behaviour.

Elastic anisotropy is directly connected to seismic anisotropy, which is a phenomenon in which the seismic wave speed varies as a function of the propagation direction. It is mainly observed in the crust, mantle boundary layers, and inner core (Almqvist and Mainprice, 2017; Karato, 1998; Kendall, 2000; Long and Becker, 2010; Deuss, 2014). Understanding and modelling seismic anisotropy is crucial for determining long-term deformational patterns and the present-day stress field in the crust, and constraining geodynamic modelling predictions (Jadamec and Billen, 2010; Hu et al., 2017; Zhou et al., 2018; Lo Bue et al., 2022). In addition, the ability to account for anisotropic effects can improve the quality of subsurface imaging. Indeed, it has been demonstrated that, because of the uneven seismic ray coverage, failing to account for seismic anisotropy may generate strong artefacts that substantially bias our understanding of mantle structures and dynamics in different tectonic settings (Bezada et al., 2016; VanderBeek and Faccenda, 2021; VanderBeek et al., 2023; Faccenda and VanderBeek, 2023). Considering the widespread presence of seismic anisotropy, anisotropic seismic models provide a more realistic representation of the Earth's subsurface compared to isotropic models.

Viscous anisotropy modelling refers to the study and simulation of materials that exhibit varying degrees of viscosity (resistance to viscous deformation) in different directions. Although viscous anisotropy has been traditionally associated with the mechanical behaviour of multi-layered media (Mühlhaus et al., 2002; Kocher et al., 2006), it has been also observed in experimentally-deformed mica-rich and olivine crystal aggregates (Shead and Kronenberg, 1993; Hansen et al., 2012). Previous numerical studies demonstrated that viscous anisotropy can potentially stabilise long-wavelength convective patterns (Christensen, 1987; Mühlhaus et al., 2004) and more generally affect processes such as plate motion (Kiraly et al., 2020), post-glacial rebound (Han and Wahr, 1997), lithospheric shear zone reactivation (Tommasi et al., 2009) and dripping (Lev and Hager, 2008).

Over the last few years a few attempts have been made to integrate micro-mechanical modelling of fabric evolution with large-scale geodynamic models using either directors (Lev et al., 2008; Halter et al., 2022), the CPO model D-REX (Kaminski et al., 2004; Becker et al., 2006; Jadamec and Billen, 2010; Faccenda and Capitanio, 2013; Faccenda, 2014; Ito et al., 2014; Hu et al., 2017; Zhou et al., 2018; Fraters and Billen, 2021), or the CPO model Visco-Plastic Self Consistent (VPSC; Tommasi et al., 2009; Li et al., 2014). However, each of these methodologies has its own limitations mainly associated with either the accuracy of the estimates, the large computational burden or software accessibility, which have impeded a more widespread diffusion in the geodynamic community. At the same time, the recovery of 3D seismic anisotropy patterns has been traditionally considered intractable due to the highly underdetermined nature of the inverse problem, and, although it has been addressed by several previous studies (e.g., Debayle et al., 2005; Panning et al., 2006; Abt and Fischer, 2008; Long et al., 2008; Wookey, 2012; Mondal and Long, 2019), only recently a theoretical background and computational algorithms have been developed to simultaneously invert for P- and/or S-wave isotropic velocity anomalies and anisotropy resulting from arbitrarily oriented structures ( Munzarova et al., 2018; VanderBeek and Faccenda, 2021; Rappisi et al., 2022; Wang and Zhao, 2022; VanderBeek et al., 2023; Del Piccolo et al., 2023). Yet, to date there is no freely-available software capable of modelling seismic anisotropy related to arbitrarily oriented structures.

In this contribution we present the new and open-source software package ECOMAN (**E**xploring the **CO**nsequences of **M**echanical **AN**isotropy) that enables linking seismology and geodynamics by providing a set of computationally optimised programs for (i) estimating rock mechanical anisotropy as a function of the geodynamic model deformation history, and compositional, rheological, stress, pressure, temperature, fields, and (ii) solving forward/inverse seismological problems accounting for seismic anisotropy. In the next sections, we first describe the different ECOMAN modules, after which we discuss the advantages and limitations of the software package, and  the roadmap for future developments.

Table 1. Abbreviations and their description. Units are indicated for dimensional physical properties.

| Abbreviation | Description | |
|---|---|---|
| | | |

| | | |
|---|---|---|
| UM | Upper mantle | |
| UTZ | Upper mantle transition zone | |
| LTZ | Lower mantle transition zone | |
| LM | Lower mantle | |
| LPO/CPO | Lattice/crystal preferred orientation | |
| SPO | Shape preferred orientation | |
| FSE | Finite Strain Ellipsoid, defined by the eigenvalues and eigenvectors of **LS** | |
| STILWE | Smoothed Transversely Isotropic Long-Wavelength Equivalent | |
| DEM | Differential Effective Medium | |
| VPSC | Visco-Plastc Self Consistent | |
| MDM | Modified Director Method | |
| | | Units |
| $\rho$ | Density | $kg/m^3$ |
| P | Pressure | Pa |
| T | Temperature | K |
| Fd | Fraction of dislocation creep deformation | - |
| **V**, $V_i$ | Velocity vector and its components | m/s |
| **F**, $F_{ij}$ | 2nd-order deformation gradient tensor and its components | - |
| **LS**, $LS_{ij}$ | 2nd-order left stretch tensor and its components | - |
| **C**, $C_{\alpha\beta}$ | 4th-order elastic tensor and its components in Voigt notation | GPa |
| **S**, $S_{\alpha\beta}$ | 4th-order compliance tensor and its components in Voigt notation | $GPa^{-1}$ |
| $\boldsymbol{\eta}$, $\eta_{\alpha\beta}$ | 4th-order normalised viscous tensor and its components in Voigt notation | - |

94

## 2 Software package structure

ECOMAN includes several programs that are complementary and can be grouped into three main categories (Fig. 1):

1) programs that estimate strain/stress-induced rock fabrics (LPO and SPO) and their elastic and viscous anisotropic mechanical properties (D-REX_S, D-REX_M, EXEV);

2) programs that post-process the simulated rock fabrics for visualisation of their isotropic/anisotropic mechanical properties and deformational history (VIZTOMO, VIZVISC), and format the elastic tensors generating input files for seismological synthetics (VIZTOMO);

3) programs that test the elastic response of anisotropic media by performing seismological forward/inverse modelling and, in particular, isotropic and anisotropic seismic tomographies on synthetic and real seismic datasets (SKS-SPLIT, PSI).

Most of the code modules are written in the Fortran programming language, except for the PSI program that is written in Julia. Visualisation of the output is done through the MATLAB MTEX toolbox (Mainprice et al., 2011) for single aggregate fabrics, and ParaView (Ahrens et al., 2005) for 2D and 3D simulations.

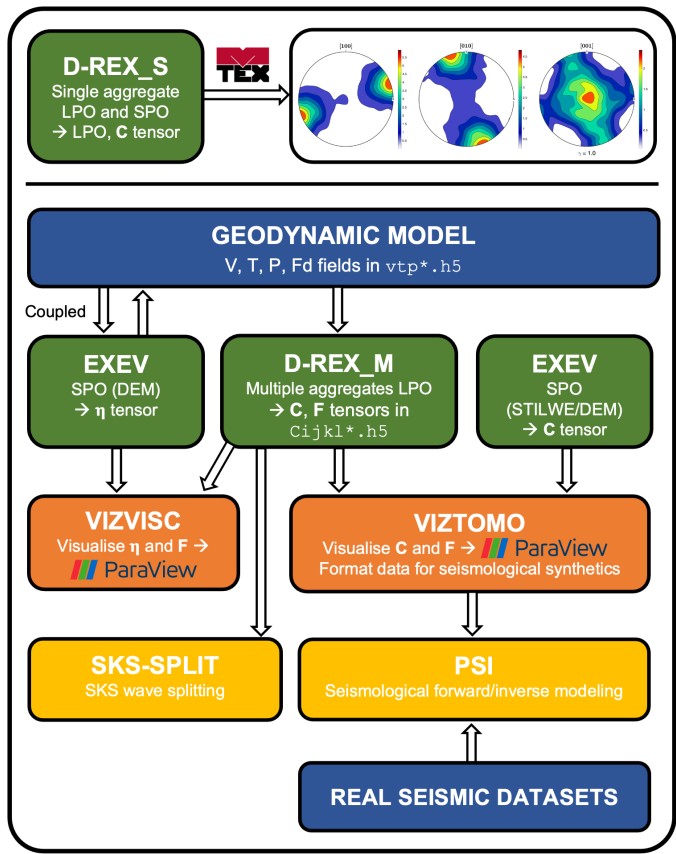

Figure 1. ECOMAN structure and flow chart. Coloured boxes denote programs that compute rock fabrics (green), post-process the elastic (**C**), viscous ($\eta$) and deformation gradient (**F**) tensors for visualisation and/or data formatting for seismological synthetics (orange), and perform seismological forward/inverse modelling on synthetic or real datasets (yellow). Input data are from geodynamic modelling or real seismic datasets (blue). Visualisation of the mechanical properties and LPO can be done with the MTEX MATLAB toolbox for single crystal aggregates or the software ParaView for large-scale simulations.

## 2.1 Rock fabrics and mechanical properties simulations

The evolution of the strain-induced LPO can be simulated with the D-REX model (Kaminski et al., 2004), which has been adapted to reproduce the fabric evolution of a single (D-REX_S) or multiple (D-REX_M) two-mineral phase aggregates representative of the entire Earth's mantle. The accuracy of the D-REX model has been tested against analytical solutions derived by Fraters and Billen (2021) (Appendix A). Five sets of two-mineral phases mantle aggregates can be defined as a function of depth or density $\rho$ (Fig. 2; see Table 1 for a list of abbreviations and physical properties):

1) **olivine** + **enstatite**, for the upper mantle (UM: 0-410 km or $3000 < \rho \leq 3650$ kg/m$^3$);
2) **wadsleyite** + majoritic garnet, for the upper mantle transition zone (UTZ: 410-520 km or $3650 < \rho \leq 3870$ kg/m$^3$);

3) ringwoodite + majoritic garnet, for the lower mantle transition zone (LTZ: 520-660 km or $3870 < \rho \leq 4150$ kg/m$^3$);

4) **bridgmanite** + ferropericlase, for the lower mantle (LM: 660-2900 km or $\rho > 4150$ kg/m$^3$);

5) **post-perovskite** + ferropericlase, for the bottom of the lower mantle according to the parametrized phase boundary

$P(GPa) = 98.7 + T(K) \cdot 0.00956$ (Oganov and Ono, 2004).

The strain-induced LPO is computed for the phases in bold, while other major phases such as garnet, ringwoodite and

ferropericlase are considered to be isotropic and their distribution is set to be random. Thus, no LPO is calculated for the LTZ,

such that (minor) anisotropy arises only when SPO modelling due to compositional layering is active (section 2.1.3; Faccenda

et al., 2019). The LPO of wadsleyite, bridgmanite and post-perovskite are computed according to the same D-REX

methodology originally applied to upper mantle phases and described in detail in Kaminski and Ribe (2001), and Kaminski et

al. (2004), but including additional slip systems as indicated in Table A1. For each of these new phases it is possible to define

in the D-REX_S and D-REX_M input files the same free parameters of the D-REX model (the dimensionless grain boundary

modility $M^*$, nucleation parameter $\lambda^*$, threshold volume fraction $\chi$, and in addition the power-law exponent for dislocation

creep deformation within the grains) as for upper mantle aggregates. The full elastic tensor is then calculated according to the

crystal orientation, volume fraction, phase abundance, P-T conditions, bulk rock composition, and using Voigt-Reuss-Hill

averaging schemes (see section 2.1.2 and Appendix B for more details).

The elastic properties related to strain/stress-induced SPO fabrics can instead be calculated at the grain- or rock-scale and for

layered or two-phase (matrix-ellipsoidal inclusions) systems using the isotropic elastic moduli of the different (fluid, mineral,

rock) components (EXEV).

The elastic properties and density of the aggregates characterised by LPO and/or SPO fabrics are estimated at relevant mantle

P-T conditions using the single crystal elastic moduli and their P-T derivatives for the main mineral phases and compiled from

different mineral physics studies, together with lookup tables of the isotropic elastic moduli, density and mineral phase volume

fraction generated with MMA_EoS (Chust et al., 2017) for five bulk rock compositions (dunite, harzburgite, pyrolite, MORB,

pyroxenite) at temperature and pressure ranges of  T = 300 : 50 : 4500 K, P = 0 : 0.1 : 140 GPa, respectively (see Appendix B

and Tables B1 and B2).

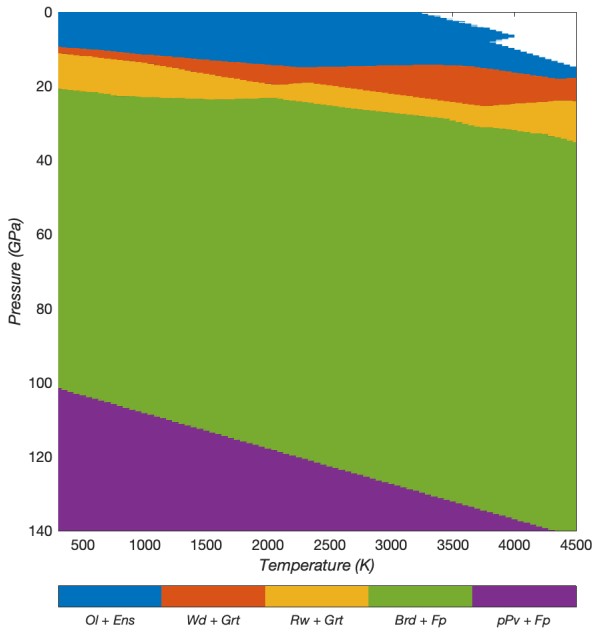

**Figure 2**. Two-phase aggregates defined by the density crossovers and Brd-pPv phase transition for a pyrolytic mantle composition.

### 2.1.1 D-REX_S

D-REX_S is a program designed for modelling the evolution of strain-induced LPO fabrics and related elastic properties of a single, two-mineral phases mantle aggregate, as a function of the imposed flow field, amount of strain, crystal plasticity, P-T conditions and additional effects related to SPO fabrics. It builds on the original D-REX software (Kaminski et al., 2004) for modelling the strain-induced LPO, and it includes MATLAB scripts to generate pole figures of the LPO and isotropic/anisotropic seismic properties with the MTEX software (Mainprice et al., 2011) (Fig. 3a-d), together with the possibility to display the evolving fabric strength (M-index, J-index) and the fraction of different anisotropy components (obtained via tensor decomposition; Browaeys et al., 2004; Fig. 3e-h).

D-REX_S is particularly useful for those users who are not familiar with LPO modelling, and more in general, to anyone interested in performing parameter sensitivity tests on different mantle mineral aggregates before launching large-scale simulations. In addition, the microstructures generated with D-REX_S can be used in the D-REX_M 2D-3D simulations to impose pre-existing (e.g., fossil) fabrics on multiple crystal aggregates located within a specific subdomain (see section 2.1.2). As an application and with the aim of debunking the common misconception that the D-REX model can only produce too strong LPO fabrics, here we show that with a suitable choice of D-REX free parameters it is possible to reproduce both weak and strong olivine fabrics as recorded in natural samples (Warren et al., 2008) and high-strain laboratory experiments (Hansen et al., 2014; Tasaka et al., 2017) (Fig. 4).

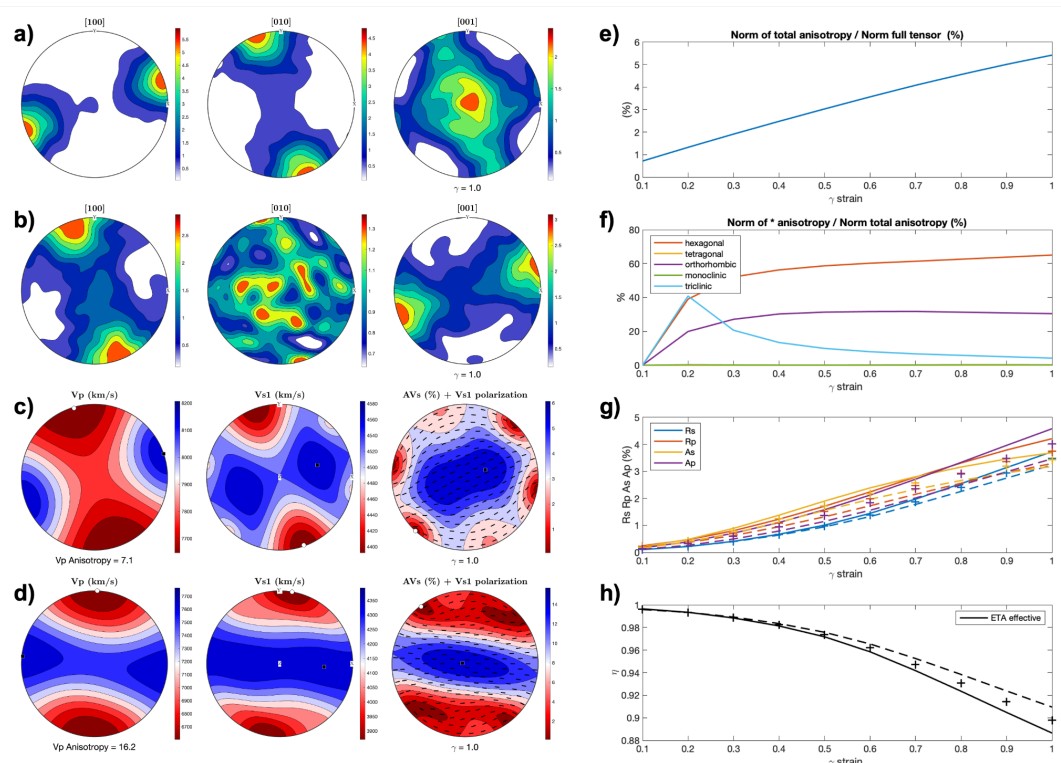

**Figure 3.** D-REX_S output for an upper mantle aggregate (Ol:Opx=70:30) subjected to a simple shear deformation of 1. Pole figures of the
olivine (a) and orthopyroxene (b) crystallographic axes; (c) pole figures of Vp, Vs1, AVs = 200(Vs1 - Vs2)/(Vs1 + Vs2) with superimposed
Vs1 polarisation directions evaluated from the elastic tensor of the two-phase aggregate. (d) same as (c) but with the superimposed effect of
an SPO fabric due to 5% melt-filled cracks aligned at -30° from the principal stress (i.e., at 15° from the horizontal plane); (e) fraction of
total anisotropy relative to the full elastic tensor and (f) contribution of 5 anisotropic classes relative to the total anisotropy; (g) P- and S-
wave radial and azimuthal anisotropy and (h) eta parameter = F/(A-2L) for the elastic tensors computed with the Voigt (continuous lines),
Reuss (dashed lines) and Hill (crosses) averaging schemes, respectively.

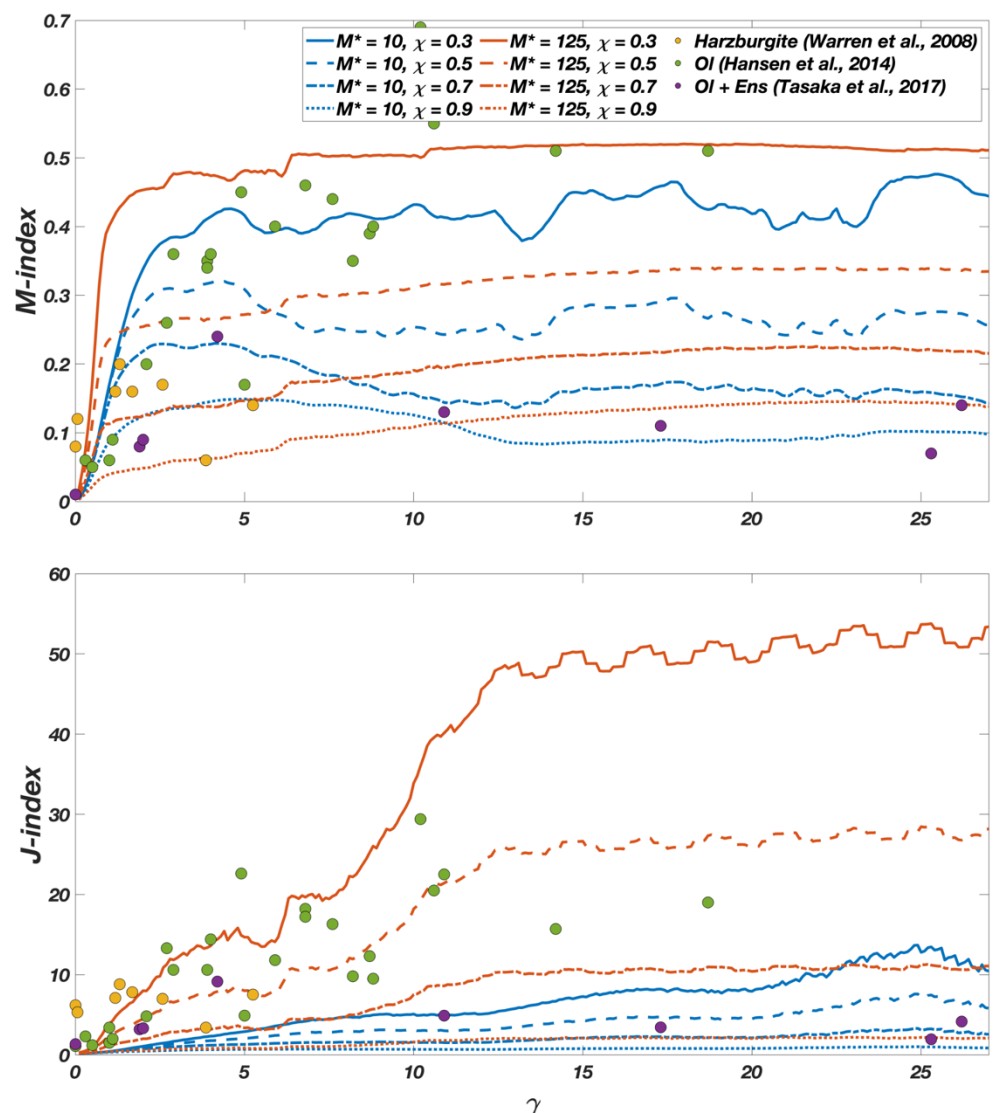

**Figure 4** –M-index (top) and J-index (bottom) for olivine fabrics developing in simple shear deformation as computed with D-REX_S using different parameters and compared with data reported in the literature as indicated in the legend. Fixed parameters are $\lambda^* = 5$, olivine $nCRSS = (1, 2, 3, \infty)$, and the aggregates have a phase abundance of $Ol: Ens = 70: 30$ with 1000 crystals for each phase.

**2.1.2 D-REX_M**

D-REX_M is a program that computes the evolution of the LPO and related elastic properties of multiple, two-mineral phases mantle aggregates, as a function of the single crystal plastic and elastic properties, and of the flow field, deformation mechanisms and P-T conditions resulting from 2D-3D geodynamic simulations. It builds on the original D-REX software, which includes routines for estimating the strain-induced LPO and elastic properties (i) for upper mantle polycrystalline

aggregates only, (ii) using single crystal elastic tensors derived at room P-T conditions and averaged using a Voigt scheme, (iii) in a 2D Cartesian domain, (iv) for a single (steady-state) flow field, and (v) whereby the whole deformation is considered to be accommodated by dislocation creep assisted by grain-boundary sliding (Kaminski et al., 2004). D-REX_M additionally models:

- fabrics relevant to the mid and lowermost mantle (Faccenda, 2014), including those with post-perovskite. Phase transitions can be set to occur at predefined depths (e.g., 410 km, 660 km), density crossovers (which allow modelling the deflection of phase boundaries with a non-zero Clapeyron-slopes; Fig. 7), and parameterized phase boundaries as for the case of post-perovskite (Oganov and Ono, 2004). When a particle enters the stability field of another anisotropic phase, the LPO can be either reset or retained (which implies axisymmetric topotactical growth);

- elastic properties and density as a function of the bulk rock composition and local P-T conditions (Faccenda, 2014; Chang et al., 2016; Ferreira et al., 2019) (Appendix B). The isotropic component of the elastic tensors and density are taken from the lookup tables generated by MMA-EoS for a given mantle lithology (Table A1), and the anisotropic component from the mineral single crystal elastic moduli and their pressure and temperature derivatives listed in Table B2. This strategy ensures a gradual transition of the seismic properties at phase boundaries where phase transformations occur. Voigt-Reuss-Hill averaging schemes of the elastic moduli are included;

- non-steady-state flows in 2D/3D Cartesian and polar grids (Faccenda and Capitanio, 2012, 2013, Hu et al., 2017; Zhou et al., 2018; Lo Bue et al., 2022; Faccenda and VanderBeek, 2023; Rappisi et al., 2024). In polar coordinates the velocity gradient tensor must be computed in the external Cartesian reference frame, as described in Appendix C. The global-scale models are spatially discretized using the so-called Yin-Yang grids (Kageyama and Sato, 2004). Several examples (cookbooks) are provided on how to use the software in different coordinate systems and in steady-state or time-dependent flow conditions;

- fabric evolution in the presence of multiple creep mechanisms. At any time step, the fraction of deformation accommodated by dislocation creep in a given point of the geodynamic model defines the fraction of time spent for intracrystalline deformation assisted by grain-boundary sliding according to the D-REX model. The remaining time is used to apply, when present, fluid deformation rotation to the whole crystal aggregate (e.g., Hedjazian et al., 2017) (Appendix D);

- a pre-existing (fossil) fabric (pre-computed with D-REX_S) within a subdomain, typically the lithosphere. This is often the case for geodynamic models where the lithosphere accretion is not modelled, and its geometry is initially prescribed.

The D-REX_M input files should contain information about the geodynamic model evolution. Critical information are the components of the velocity vector **V** field, and, for time-dependent flow models, the total elapsed time and timestep. These are used to compute the velocity gradient tensor, and the LPO evolution and advection of the crystal aggregates. Additional fields defined on the Cartesian/spherical grid that can be included in the input files are:

- when the geodynamic model is thermo-mechanical, the temperature T and total pressure P fields, which can be used to compute the single crystal elastic tensor and aggregate phase transitions as a function of the local P-T conditions;
- when the rheological model of the geodynamic simulation is based on multiple visco-plastic deformation mechanisms, the fraction of deformation accommodated by dislocation creep $F_d = \eta_{disl}/\eta_{eff}$, where $0 \leq F_d \leq 1$, $\eta_{disl}$ is the viscosity calculated with the dislocation creep flow law, $\eta_{eff}$ is the effective viscosity calculated with the harmonic average of each of the viscosities representing a different deformation mechanism.

In summary, the time and the velocity field **V** are essential information, while the P, T, Fd fields are optional and depend on the type of (mechanical vs. thermomechanical) geodynamic and (single vs. multiple visco-plastic deformation mechanisms) rheological models.

While the **V**, P, T, Fd fields are defined on the Eulerian grid, the distribution, size, modal composition and mechanical properties of mantle aggregates are defined on the Lagrangian particles. After initialising the Eulerian grid and Lagrangian particles, the entire run consists of three main steps: (i) backward advection of the particles for a given time span or up to a maximum cumulated strain; (ii) forward advection and update of the LPO and deformation gradient tensor **F**; (iii) computation of the full elastic tensor and creation of the output file. This strategy ensures a homogeneous final distribution of the aggregates (coincident with the initial one) which is desired for visualization of the aggregates mechanical and physical properties and for running seismological synthetics. The D-REX_M output file(s) includes, for each mineral aggregate, the elastic tensor **C**, density, and the deformation gradient tensor **F**. These properties can be then processed and visualised with the software VIZTOMO (section 2.2.1).

D-REX_M is parallelized using a hybrid MPI and OpenMP scheme to take advantage of multiple CPUs architectures of modern HPC clusters. Runtime is obviously affected by the number of crystal aggregates, number of crystals per aggregate and of active slip systems per crystal, and number of timesteps (Fig. 5, top). The parallel efficiency is 70-90%, and the update of the LPO and **F** tensor during forward advection is the most time-consuming part of the run (Fig. 5, centre, bottom). The small performance degradation is due to the initialization of the Eulerian/Lagrangian grids and arrays, and to I/O operations which are executed serially within each process. As a result, the efficiency of the time-dependent flow models is lower than that of steady-state models, as the latter only require a single velocity (and P, T, Fd) field to be loaded and processed.

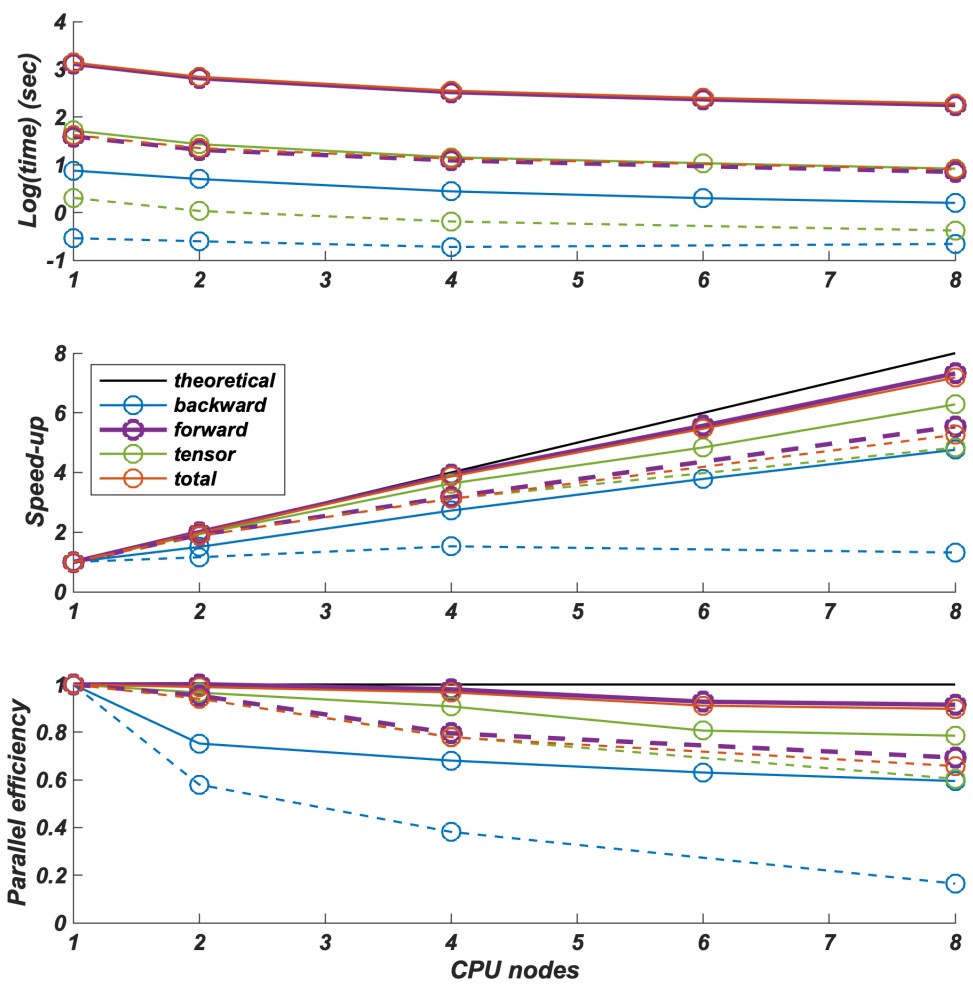

**Figure 5.** Runtime (top), speedup (centre) and parallel efficiency (bottom) of D-REX_M for initial backward advection of aggregates ("backward"), forward advection and updating of the LPO and **F** tensor ("forward"), full elastic tensor computation and output file creation ("tensor"), and entire run ("total"). Results are shown for two models included in the cookbooks: the 3Dspherical_global model (steady-state flow, 96 timesteps, 1327606 aggregates, LPO computed only for 260474 upper mantle aggregates) and the 3Dspherical_sinkingslab model (dashed lines; time-dependent flow, 20 timesteps, 38509 aggregates, LPO computed for 25177 upper mantle and 6587 upper mantle transition zone aggregates). Runs performed on a HPE Superdome Flex (8 CPUs, 28 cores Intel Xeon(R) PLATINUM 8180 @ 2.50GHz) distributing the computational load up to 8 CPU nodes, each one working independently with 28 cores.

### 2.1.3 EXEV

EXEV includes routines to compute the EXtrinsic Elastic and Viscous anisotropy using Effective Medium Theory modelling for a multi-component layered system (Smoothed Transversely Isotropic Long-Wavelength Equivalent, STILWE; Backus, 1962) or a two-component system with similar ellipsoidal inclusions in a uniform background matrix (Differential Effective Medium, DEM; e.g., Mainprice, 1997).

The elastic tensor **C** due to SPO fabrics can be either estimated independently or, when using the DEM approach, superimposed

on that obtained from the strain-induced LPO modelling. SPO fabrics that can be modelled are those related to rock- or grain-

scale layering (e.g., Faccenda et al., 2019), or to the presence of preferentially aligned ellipsoidal inclusions (e.g., melt-/fluid-

filled cracks). The user then needs to specify:

- for grain-scale layered fabrics, a dominant ultramafic or mafic lithology. In this case the mineral phase proportions
- from the MMa-EoS lookup tables define the mixture for the STILWE model;
- for rock-scale layered fabrics, the relative abundance of the five available ultramafic-mafic lithologies (dunite,
- harzburgite, pyrolite, basalt and pyroxenite) defining the mixture for the STILWE model;
- for matrix-inclusion fabrics, the elastic tensors of the two components and the inclusion's shape and volume fraction
- as required by the DEM modelling. The matrix elastic tensor can be replaced with that from the LPO modelling in
- order to estimate the combined effect of LPO and SPO fabrics.

The SPO fabrics can then be oriented at any angle relative to the principal Finite Strain Ellipsoid (FSE) axis or, in case of

cracks, the local principal stress obtained from the "present-day" (i.e., last) velocity field.

Summarising, SPO fabric modelling requires one or more of the following: **F** and/or **C** obtained from D-REX_M; P, T and/or

**V** and/or fluid/melt fraction fields for the "present-day" state of the geodynamic model. Consequently, this modelling is

performed when post-processing the D-REX_M output with the software VIZTOMO (see section 2.2.1).

The total or deviatoric component of the viscous tensor $\boldsymbol{\eta}$ due to SPO fabrics is estimated for two-phase systems with ellipsoidal

weak/hard inclusions using the DEM theory, and the parametrization of the viscous tensor evolution and orientation as a

function of the cumulated deformation (**F**) obtained following de Montserrat et al. (2022). Indeed, most if not all, mantle levels

are composed of two main mineral phases that control both the elastic and viscous properties. The case of a multi-component

layered medium is not considered because its viscous tensor can be either approximated with flat inclusions, or more simply

computed using the Voigt and Reuss averages of the layers' isotropic viscosity.

The first modelling phase requires running subprogram DEMviscous to generate a database of viscous tensors for a range of

inclusion shapes and volume fractions, and inclusion-matrix viscosity contrasts. The latter implies that the viscous moduli are

dimensionless  and can therefore be interpreted as scaling factors with respect to an isotropic effective viscosity of the bulk

rock or most abundant mineral phase. Subsequently, the database can be exploited by large-scale geodynamic simulations to

either (i) return the viscosity tensor $\boldsymbol{\eta}$ from a look-up table  for every point of the computational domain, which can be

superimposed on the isotropic effective viscosity computed from flow laws (coupled mechanical simulations), or (ii) estimate

$\boldsymbol{\eta}$ (uncoupled mechanical simulation) and/or visualise its anisotropic viscous properties for the "present-day" state of the

geodynamic model with the software VIZVISC (section 2.2.2).

## 2.2 Visualisation of the mechanical properties and data formatting in preparation for seismological synthetics

### 2.2.1 VIZTOMO

VIZTOMO processes the D-REX_M output for the visualisation of the aggregates' elastic and deformational history properties. Estimation of extrinsic anisotropy effects via the EXEV routines is possible at this stage. The properties of the **C** and **F** tensors can be determined either at the position of the Lagrangian mantle aggregates, or, in case of the elastic properties and density, interpolated to a structured (tomographic) grid. In the latter case the grid can be saved in a format suitable for generating synthetic seismic datasets via the PSI_D package (see section 2.3.2) or for 3D waveform simulations in SPECFEM (Komatitsch and Tromp, 1999).

Several properties of the elastic tensor **C** can be visualised:

- Isotropic or ray-path dependent velocity anomalies and anisotropic elastic properties of body-waves (i.e., P-wave anisotropy and direction of maximum P-wave velocity; direction and magnitude of maximum S-wave splitting delay time; S-wave radial and azimuthal anisotropy);
- reflection/transmission energies resulting from the whole range of P-S conversions occurring at discontinuities (useful for studies based on receiver function analysis);
- the fraction of the elastic tensor anisotropic component relative to the total and the relative contributions of five different anisotropy classes (hexagonal, orthorhombic, tetragonal, monoclinic, triclinic) obtained through elastic tensor decomposition (Browaeys and Chevrot, 2004);
- the orientation of the hexagonal symmetry axis (already present in the original D-REX).

The deformation history stored in **F** can be visualised in terms of the FSE shape and orientation and/or length or orientation of its minimum and maximum semi-axes.

The different fields are saved in specific file formats which can be imported by the open source ParaView software (Ahrens et al., 2005) for visualisation. Figures 6, 7, 8, 9 display some of these fields computed for different geodynamic models.

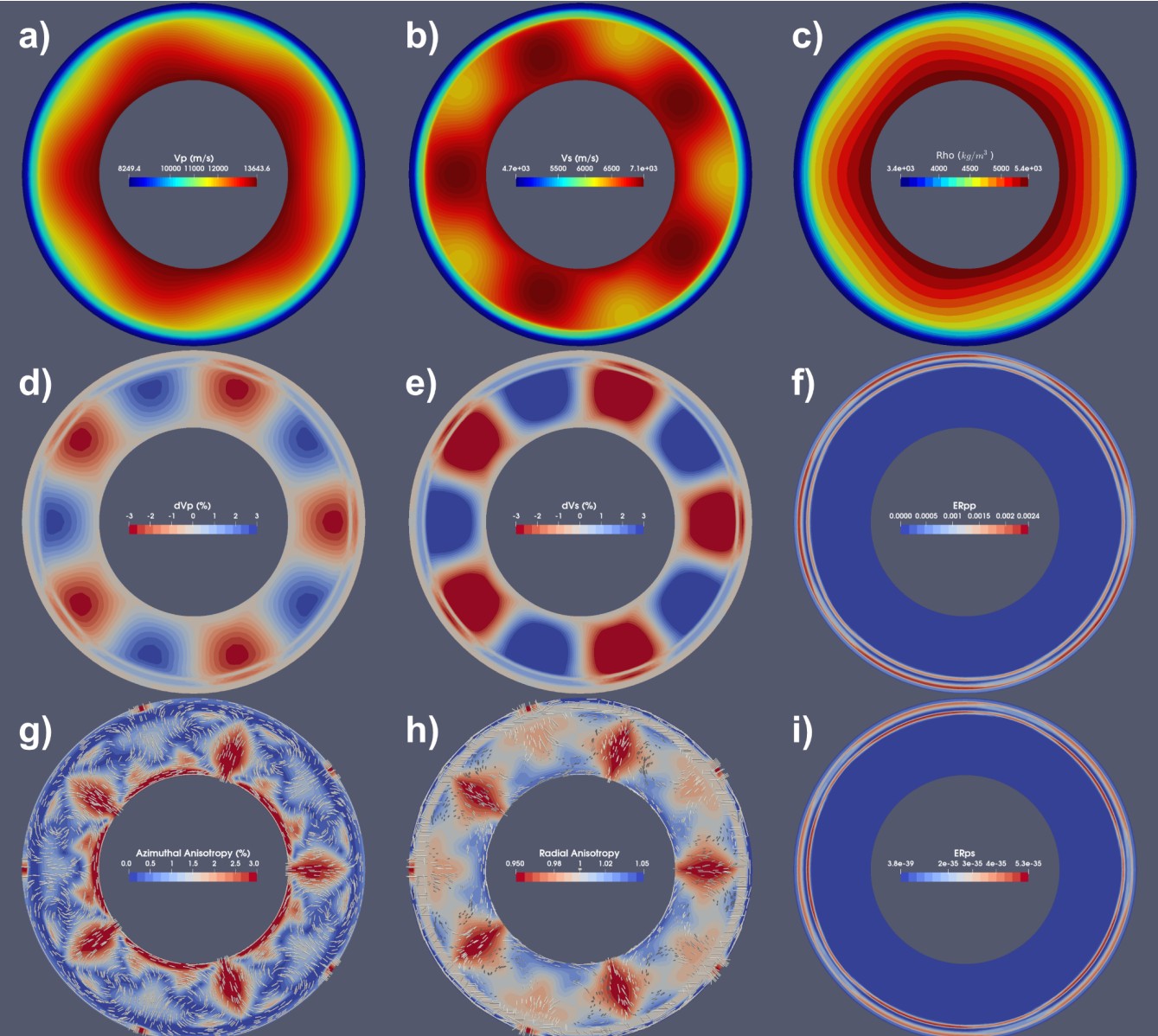

**Figure 6.** VIZTOMO output for the model 2Dpolar_convection available in the cookbooks. The steady-state 2D incompressible flow field in cylindrical coordinates is the result of the analytical solution shown in section 9.3 of the software manual and coded in the MATLAB script cookbooks/2Dpolar_convection/polarcell.m. (a,b) isotropic Vp and Vs (m/s); (c) density); (d,e) isotropic P- and S-wave anomalies; (g) azimuthal anisotropy and FSE semi-axis (white bars); (h) radial anisotropy and TTI axis (white bars); (f,i) P-P and P-S reflection energy for waves propagating upward.

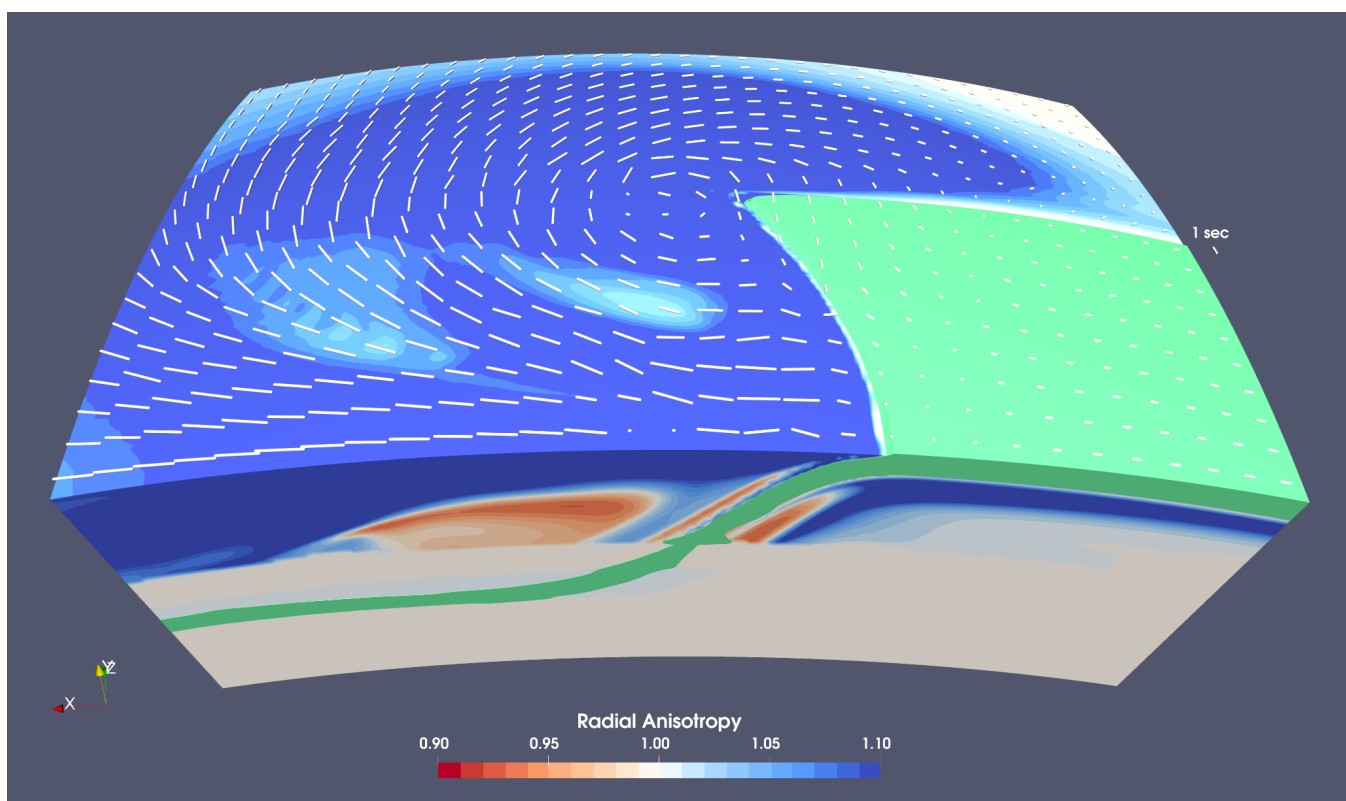

**Figure 7.** VIZTOMO output for a 3D model of oceanic plate subduction and roll back in spherical coordinates. The time-dependent flow field has been computed with the software I3MG modified to account for spherical coordinates (see Faccenda and VanderBeek, 2023). Fabrics computed with D-REX_M for only upper mantle aggregates. A fossil A-type olivine fabric with the fast axis parallel to plate motion and computed with D-REX_S is initially defined within the oceanic plate volume. The colour scale indicates radial anisotropy in the upper mantle, and the white bars the SKS splitting computed with SKS-SPLIT. The volume in green encloses material with a +2% P-wave anomaly (i.e., the oceanic plate). Note the apparently thicker slab portion around the 410 km depth discontinuity, due to the upwelling of the olivine-spinel phase transition. The model domain extends from 0-1000 km along the radial direction, 85°-115° along longitude, and 70-90° along colatitude.

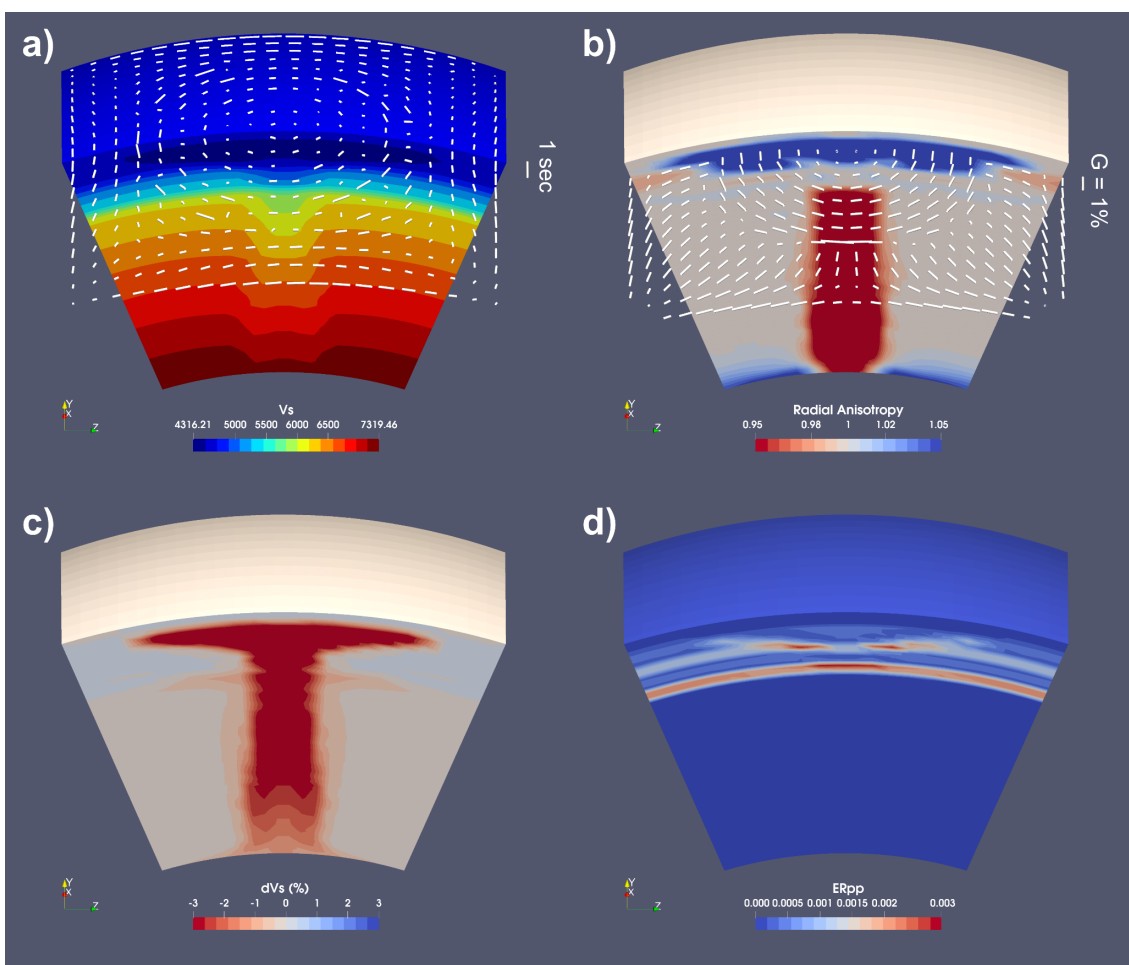

**Figure 8.** VIZTOMO output for an upwelling plume toy model in spherical coordinates. The time-dependent flow field has been computed with the software I3MG modified to account for spherical coordinates (see Faccenda and VanderBeek, 2023). a) absolute Vs (m/s) and SKS splitting computed with SKS-SPLIT (white bars); b) radial anisotropy (colour scale) and azimuthal anisotropy at 200 km depth (white bars); c) isotropic Vs anomaly; d) P-P reflection energy for a wave propagating upward. The model domain extends from 0-2900 km along the radial direction, 70°-110° along longitude, and 70-110° along colatitude.

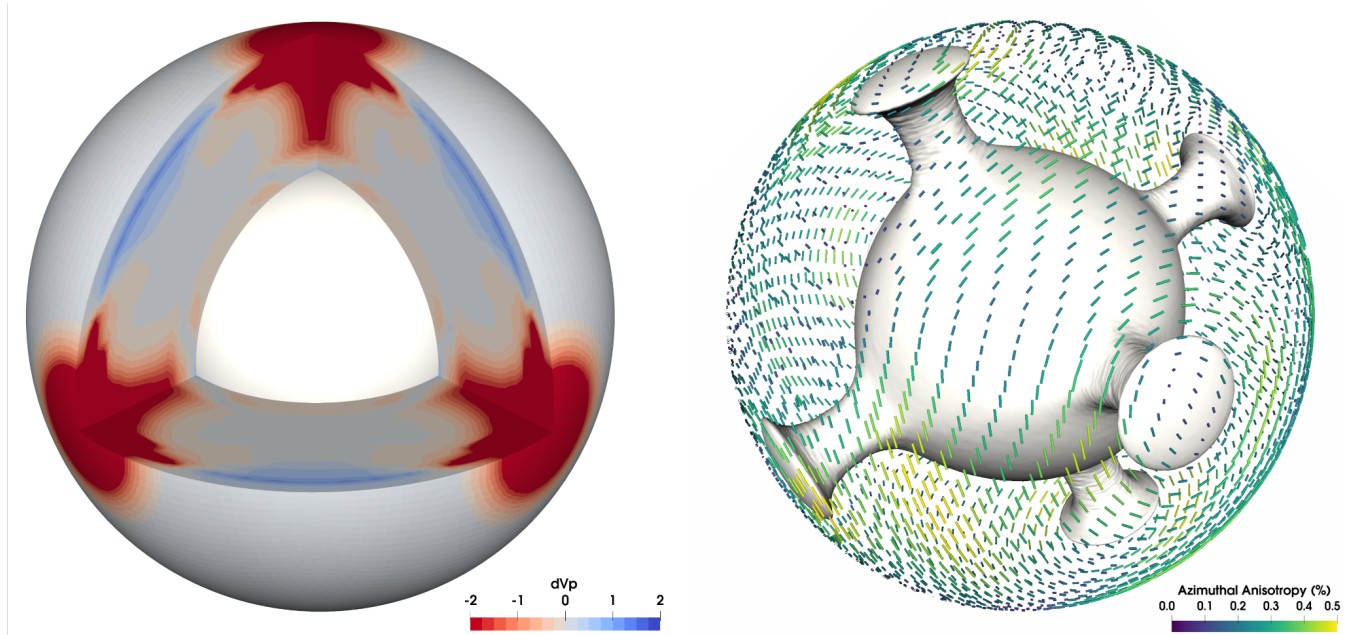

**Figure 9:** VIZTOMO output for the model 3Dspherical_global available in the cookbooks. The time-dependent flow field has been computed
with the software I3MG modified to account for 3D global spherical coordinates using the Yin-Yang grids (Kageyama and Sato, 2004).
Isotropic P-wave anomaly (left) and azimuthal anisotropy at 200 km depth (right). The grey surface encloses material 0.1 dimensionless
units hotter than the average temperature at any depth.

### 2.2.1 VIZVISC

VIZVISC processes the D-REX_M output for the visualisation of the aggregate properties such as the viscous anisotropy and
related deformational history (in terms of the FSE) in ParaView. The deviation from isotropy is evaluated by computing the
radial and azimuthal components of viscous anisotropy in a similar way as for the elastic tensor. More in detail, radial viscous
anisotropy is defined as $\xi = N/L$, while azimuthal viscous anisotropy is defined by the magnitude $G = \sqrt{Gc^2 + Gs^2}$ and
azimuth $\phi = tan^{-1}(Gs, Gc)$, where $N = \frac{1}{8}(\eta_{11} + \eta_{22}) - \frac{1}{4}\eta_{12} + \frac{1}{2}\eta_{66}$ , $L = \frac{1}{2}(\eta_{44} + \eta_{55})$, $Gc = \frac{1}{2}(\eta_{55} - \eta_{44})$, $Gs = \eta_{45}$. Radial
and azimuthal viscous anisotropy are evaluated in the FSE (and thus inclusions) reference frame, whereby the minor semi axis
is oriented along the vertical direction, and the intermediate and major semi axes are in the horizontal plane. As such, radial
anisotropy is always $\geq 1$. Figure 10 displays some of these fields computed for a 2D steady-state model of mantle convection
with periodic upwellings and downwellings.

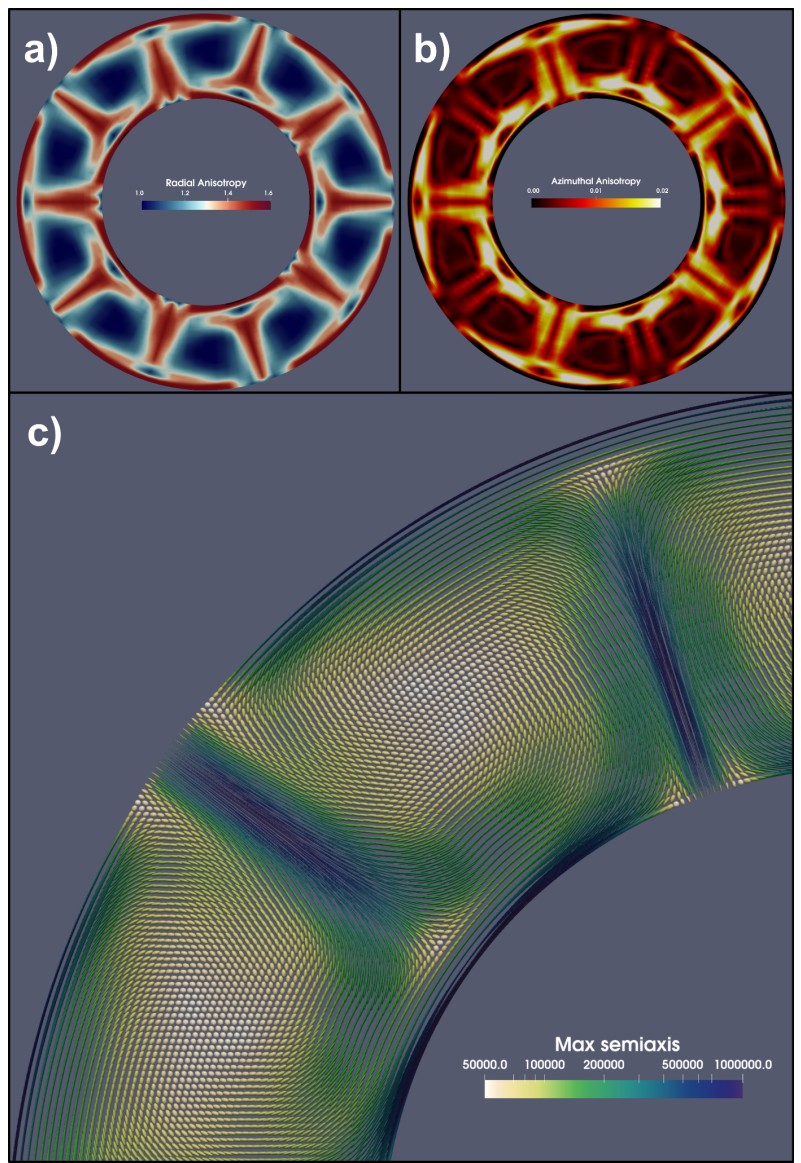

56

**Figure 10.** VIZVISC output for the model 2Dpolar_convection available in the cookbooks. The steady-state 2D incompressible flow field
in cylindrical coordinates is the result of the analytical solution shown in section 9.3 of the software manual and coded in the MATLAB
script cookbooks/2Dpolar_convection/polarcell.m. (a) Radial viscous anisotropy; (b) azimuthal viscous anisotropy; (c) FSE shape coloured
by the length of its major semi-axis upscaled by a factor of 50 km.

61

## 2.3 Seismic forward and inverse modelling

### 2.3.1 SKS-SPLIT

SKS-SPLIT estimates the SKS splitting at a grid of virtual seismic stations placed at the top of the D-REX_M model as a function of the back-azimuth using the Fortran routines included in FSTRACK (Schulte-Pelkum and Blackmann, 2003; Becker et al., 2006). These routines use the reflectivity method (for details, see Booth and Crampin, 1983; Chapman and Shearer, 1989) to compute synthetic seismograms through layered anisotropic media traversed by an incident plane wave (5° for typical SKS arrivals) over a range of frequencies (0-25 Hz). Band-pass filters from 0.1 to 0.3 Hz are then applied to construct synthetic seismograms in the SKS band (3.3 to 10s). Successively, the splitting is determined with the cross-correlation method of Menke and Levin (2003). The routines have been adapted to load the D-REX_M output, stack the elastic tensors and densities in an upper mantle rock column beneath each virtual seismic station (Faccenda and Capitanio, 2013), and run in parallel using MPI. The averaged fast azimuth scaled by the delay time can then be visualised in ParaView as shown in Figures 7 and 8a.

### 2.3.2 PSI

PSI (Platform for Seismic Imaging) is a Julia package for performing tomographic inversion of both real and synthetic seismic datasets. In the context of the ECOMAN software, it's a useful tool for exploring how features in high-resolution geodynamic simulations are mapped into lower resolution seismic tomography models; this is a critical step if one aims to evaluate geodynamic results against existing tomographic images.

PSI can be used to forward model P and S travel-times, splitting intensity (Chevrot, 2000), and shear wave splitting parameters (i.e. the fast-polarisation azimuth and the delay time between the fast- and slow-polarised waveforms). Forward modelling of these seismic observables is supported for three different model parameterizations: (1) isotropic P- and S-wavespeeds, (2) hexagonally anisotropic media defined by the 5 Thomsen parameters and the azimuth and elevation of the symmetry axis, and (3) fully anisotropic models defined by the density-normalised 21-component elastic tensor which can be generated from D-REX_M + VIZTOMO (sections 2.1.2 and 2.2.1). Seismic phase velocities are computed following Thomsen (1986) for hexagonally anisotropic models while the Christoffel equations are solved when the elastic tensor parameterization is used. Anisotropic travel-times and splitting intensities for arbitrarily polarised S-waves are computed following VanderBeek et al. (2023) using a long-wavelength approximation in which the accumulated delay time between fast- and slow-polarised qS-waves is less than their dominant period. Lastly, splitting parameters are predicted using the matrix propagation method of Rumpker and Silver (1998). In this initial release of PSI, all predictions are made via integration along ray paths traced through a user-defined 1D reference velocity model using the TauP Toolkit (Crotwell et al. 1998). The 3D model properties are subsequently interpolated to these paths before computing the seismic observables. We anticipate releasing an update to the package that includes both 3D anisotropic ray tracing and finite-frequency kernels that are currently under development.

Travel-time and splitting intensity datasets can be inverted individually and jointly either for isotropic (Vp and Vs) or hexagonally anisotropic model parameters. Hexagonal anisotropy is defined by up to five free parameters, the isotropic (1) P-

and (2) S-velocity, (3) anisotropic magnitude, and the (4) azimuth and (5) elevation of the hexagonal symmetry axis. Only a single anisotropic magnitude parameter is required because the strength of P and S anisotropy is strongly correlated (e.g., Becker et al., 2006). Consequently, the ratios between the three Thomsen parameters ($\boldsymbol{\varepsilon}$, $\boldsymbol{\delta}$, $\boldsymbol{\gamma}$; Thomsen 1986) and the inverted anisotropic magnitude parameter must be chosen *a priori* and can be spatially variable. Source and receiver statics for each observation and seismic phase type may also be included as free parameters. The tomographic model is obtained by iteratively solving a system of linearized equations relating the perturbations in the inversion parameters to the data residuals augmented with damping and smoothing constraints. The solution is obtained via the LSQR algorithm (Paige & Saunders, 1982). Full details on the tomographic method can be found in VanderBeek and Faccenda (2021) and VanderBeek et al. (2023). The final tomographic solution is written to a VTK file to be visualised in ParaView. Tomographic inversions can be run from a workstation. For the problem shown in Figure 11 and consisting of 12,320 observations and 415,044 free parameters, solutions can be obtained within ~10 minutes (less for isotropic inversions) using 6 cores and 1.8 GB peak RAM.

The PSI inversion methodology has been tested on synthetic models of oceanic plate subduction, intra-oceanic upwelling plume, spreading oceanic ridge, and Central-Eastern Mediterranean subduction (VanderBeek and Faccenda, 2021; Lo Bue et al., 2022, VanderBeek et al., 2023; Faccenda and VanderBeek, 2023), and applied to the isotropic and anisotropic imaging of the Central Mediterranean (Rappisi et al., 2022) and of the Mt. Etna volcanic field (Lo Bue et al., 2024). In Figure 11, we illustrate results obtained from a synthetic inversion of direct teleseismic P- and S-wave relative travel-times computed through the subduction zone shown in Figure 7. Synthetic data were computed using the full 21-component elastic tensor while the inversion was performed for the best-fit hexagonal anisotropic parameters. We considered an array of 770 receivers extending from ±7.5° in longitude and ±11.5° in latitude (~75 km spacing) that recorded 16 events; 8 at a range 50° and another 8 at 80° from the origin of the subduction zone model and equally distributed in back-azimuth. This example is included in the PSI package.

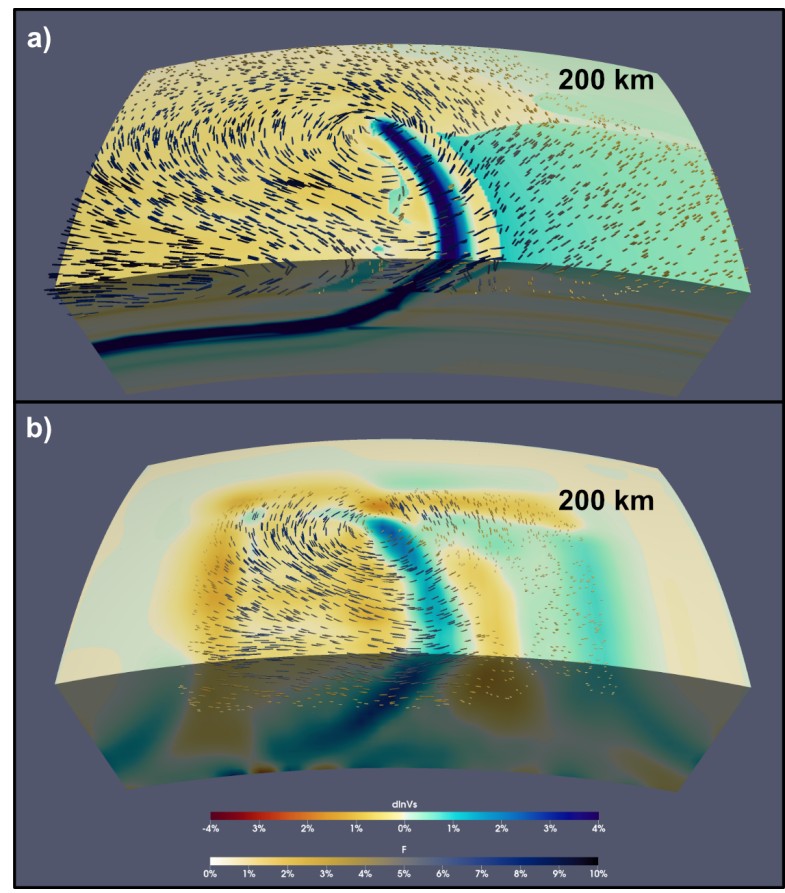

**Figure 11.** Synthetic tomography results obtained from PSI for the model shown in Fig. 7. (a) Target anisotropic model generated from VIZTOMO. (b) Recovered anisotropic model obtained by inverting synthetic teleseismic P and S (relative) delay times computed from the model in (a). In both panels, the isotropic S-wave velocity perturbations are computed with respect to the far-field 1D velocity profile. Quivers parallel the hexagonal symmetry axis and are scaled and coloured by the anisotropic strength. The top surface shown is located at 200 km depth while the full model extends from 0-1000 km along the radial direction, 85°-115° along longitude, and 0-20° along latitude.

## 3 Discussion

### 3.1 Software advantages

When compared to other similar software, ECOMAN (i) is a more versatile package suitable for any geodynamic simulations (2D and 3D; Cartesian and polar coordinate systems; regional and global settings), (ii) allows for the possibility to account for the time-dependent deformational history of the mantle (which is usually not steady-state, especially close to plate boundaries and hotspot settings), (iii) computes the strain-induced fabric and elastic tensor of different mantle layers (i.e., not only the upper mantle), (iv) includes Effective Medium Theories (STILWE, DEM) and a parametrization of the fabric evolution of two-phase composites to predict elastic and viscous extrinsic anisotropy, (v) generates realistic grid structure distributions of mantle elastic properties to be used for forward/inverse seismological modelling (e.g., PSI, SPECFEM3D), and (vi) performs synthetic seismic inversions (e.g., P- and S-wave travel-time tomographies, S-wave splitting intensities) within the

computational domain, which facilitates the comparison with other tomographic models and the estimation of apparent anomalies (artefacts) due to, for example, unaccounted-for elastic anisotropy (Bezada et al., 2016; VanderBeek and Faccenda, 2021; VanderBeek et al., 2023) and/or regularisation.

## 3.2 Software (real or potential) current limitations

D-REX_S and D-REX_M compute the strain-induced LPO for the most abundant and highly anisotropic mineral phases, while assuming random orientation for several secondary phases that instead could contribute substantially to the aggregate anisotropic properties. For example, cubic ferropericlase is relatively abundant (< 20%) and becomes highly anisotropic in the lower half of the lower mantle (Avs = 30-50%), such that it could dominate seismic anisotropy at these depths (Marquardt et al., 2009). Davemaoite (Ca-perovskite) is also a highly anisotropic mineral with cubic symmetry (AVs = 25-15%, Kawai and Tsuchiya, 2015) but its model abundance is quite low (< 10%). Recently, micromechanical simulations of strain-induced LPO in aggregates with a pyrolite mantle composition have shown that the bulk aggregate seismic anisotropy is controlled by bridgmanite and post-perovskite, while the cubic secondary phases appear to only slightly reduce the amplitude of anisotropy (Chendler et al., 2021). The latter effect can thus be well approximated by randomising the orientation of the cubic phases as assumed here.

A main limitation of ECOMAN is that it does not include yet the modelling of viscous anisotropy due to the intrinsic mechanical anisotropy of crystals. Tommasi et al. (2009), have used VPSC simulations to show that olivine CPO in the lithosphere can control the reactivation of fossil faults misoriented with respect to the stress field. Kiraly et al (2020), employed the modified director method (MDM) and estimated up to 1 order of magnitude of intrinsic viscous anisotropy in olivine aggregates, which can control the kinematics and dynamics of tectonic plates. Both the VPSC and MDM approaches are computationally expensive and appear to be prohibitive for the number of mantle aggregates required to discretize the mantle domain of large-scale 3D simulations. An alternative approach which minimises the computational time is therefore desired.

Fraters and Billen (2021) have released a version of the D-REX model that is embedded into the ASPECT geodynamic modelling software (Bangerth et al., 2020) and that takes into account changes in olivine fabric as a function of the water content and deviatoric stress (Karato et al., 2008). Although at present D-REX_M only allows the definition of a single olivine fabric per run, changes in crystal fabrics can be easily implemented in the future by loading the water content and deviatoric stress field from the geodynamic models and by parametrizing the boundaries of the olivine fabric domains.

Tape and Tape (2024) recently reformulated the elastic decomposition method proposed by Browaeys and Chevrot (2004) that they argued is not entirely accurate. Although this will only affect the visualization of the tensor components as performed by D-REX_S and VIZTOMO, it is planned to replace the existing elastic tensor decomposition methodology with the one proposed by Tape and Tape (2024).

A minor limitation is that the different code modules are based on different programming languages (Fortran, Julia), libraries (OpenMP, MPI, HDF5) and software (MATLAB, ParaView), whose installation on local devices might discourage potential

users. There are a few main reasons for this. First of all, the original D-REX software was written in Fortran, thus its modifications into D-REX_S and D-REX_M software was more straightforward by maintaining the same programming language. Fortran has high performance standards on HPC clusters, often if not always higher than other interpreted programming languages (e.g., MATLAB or Python). Given the large-scale computational power needed for the D-REX_M simulations, especially those with 100,000s or millions of crystal aggregates, and considering that the required compilers and libraries are routinely installed in most (if not all) HPC clusters, the usage of the ECOMAN's Fortran-based applications in high-performance environments is warranted. In contrast, the visualisation of the Fortran-based applications' output in MATLAB and ParaView can be performed on local devices. In particular, the MTEX software is a MATLAB toolbox for visualisation of the LPO and elastic tensor pole figures which should be downloaded (https://mtex-toolbox.github.io/download) and installed along with MATLAB only when using D_REX_S. Seismic forward/inverse modelling with PSI can be performed on either a HPC cluster or local devices upon installation of the Julia package. Julia was chosen because it is an open-source and high-level language that offers a number of performance benefits over other popular scientific computing languages such as Python or Matlab. It is important to stress that for any of these applications the user only needs to modify input text files, and thus no particular prerequisite or computational skill is required.

Finally, running D-REX_M requires allocating ~ 160 GB of memory per million of crystal aggregates with 1000 x 2 crystals each. This potential problem can be addressed by distributing the computational and memory load over several nodes, which is possible thanks to the hybrid MPI and OpenMP parallelization scheme.

## 4 Conclusions and outlook

ECOMAN is an open-source software package for estimating strain/stress-induced fabrics in mantle aggregates, their mechanical properties, and how mechanical anisotropy affects the geodynamic evolution and seismic imaging of the Earth's interior. Programs included in ECOMAN are portable across different HPC and local device systems (provided the Julia package and Fortran compilers are available), and are applicable to any 2D-3D geodynamic simulation. Computationally expensive programs such as D-REX_M are parallelized, offering a nearly perfect scaling with an increasing number of cores. As a result, the strain-induced fabrics of millions of mantle aggregates can now be estimated with a reasonable amount of time and computational resources.

As ongoing developments, we are seeking to include in ECOMAN micromechanical modelling methods that are capable of estimating the strain-induced LPO and/or the intrinsic viscous anisotropy at computational speeds that are orders of magnitude faster than current ones. For instance, Ribe et al. (2019), have proposed an analytical finite-strain parameterization for texture evolution in deforming olivine polycrystals that is $\approx 10^7$ times faster than full homogenization approaches such as the second-order self-consistent model. When implemented in ECOMAN, preliminary tests indicate that this new method outperforms D-REX by 1-2 orders of magnitude (Ribe et al., 2023).

In addition, in the near future the PSI software will be updated to include trans-dimensional Bayesian Monte Carlo sampling methods that, in contrast to deterministic approaches, address the consequences of under-determination in seismic imaging by constraining the uncertainty (Del Piccolo et al., 2023). Isotropic and anisotropic seismic imaging with PSI is currently feasible using body wave information such as travel-times and S-wave splitting intensity. However, when local deep seismicity is absent, as is the case in warm subduction zones or at spreading ridges and intraplate settings, the retrieved isotropic and anisotropic mantle structures are only partially recovered and often affected by smearing (e.g., Faccenda and VanderBeek, 2023). Consequently, we are planning to complement body wave information with surface wave data to improve the seismic ray coverage and the resolving power of tomographic models. Lastly, to improve the prediction of seismic observables, 3D anisotropic ray tracing and finite-frequency kernels are planned for a future release.

**Code availability**

The version of ECOMAN described in this article is 2.0 and is freely available at github (https://github.com/ecoman-geos) and zenodo (ECOMAN2.0-geodynamics: https://doi.org/10.5281/zenodo.10599735; ECOMAN2.0-seismology.SKS-split: https://doi.org/10.5281/zenodo.11173260; ECOMAN2.0-seismology.PSI_D: https://doi.org/10.5281/zenodo.11186805 ).

**Data availability**

The software package contains the input files to generate the synthetic models and datasets discussed in this manuscript. No data has been produced for this work.

**Author contributions**

MF and BPV developed the software package, with important contributions from AdM, JY and FR. MF wrote the manuscript draft. MF acquired the funding supporting the research activities. All authors have contributed to the discussion and manuscript editing.

**Competing interests**

The authors declare that they have no conflict of interest.

## 16 Acknowledgements

MF is indebted to F.A. Capitanio for supporting the beginning of this journey, T.W. Becker for providing routines to compute shear wave splitting, E. Kaminski for fruitful discussions on strain-induced LPO modelling, and to A.M. Ferreira for the continuous and stimulating feedback. Two anonymous reviewers helped in improving the article.

## 20 Financial statement

This study is supported by the ERC StG 758199 NEWTON.

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

**Appendix A. D-REX model vs analytical solution**

We compare the numerical solution of the D-REX model included in D-REX_S and D-REX_M with the analytical solution provided by Fraters and Billen (2021, Supplementary Information) for a test with an upper mantle aggregate ($Ol: Ens = 50: 50$) composed by 5 crystals for each phase, $M^* = 50$, $\chi = 0.3$, and $nCRSS = (1, 2, 3, \infty)$ for the four olivine slip systems $[100](010), [100](001), [001](010), [001](100)$ (Table A1). For the first crystal, the time derivative of the cosine direction matrix (CDM) $\frac{\delta a_{ij}^v}{\delta t} = 0$ and the vorticity matrix $\bar{\omega}_{ij}^v = \varepsilon_{jki}\omega_k^v = 0$, while for the other 4 crystals:

$$\frac{\delta a_{ij}^v}{\delta t} = \begin{bmatrix} -0.0040153 & -0.0010038 & 0.0020076 \\ -0.0085325 & -0.0010038 & 0.0065249 \\ -0.0130497 & -0.0010038 & 0.0110421 \end{bmatrix} \tag{A1}$$

with a maximum error of $3.1 \cdot 10^{-3}$ % which we attribute to truncation errors, and:

$$\bar{\omega}_{ij}^v = \begin{bmatrix} 0 & 0.005019 & 0.010038 \\ -0.005019 & 0 & 0.005019 \\ -0.010038 & -0.005019 & 0 \end{bmatrix} \tag{A2}$$

The changes in volume fractions are $\frac{\delta f_m^v}{\delta t} = 0.630116863029$ for grain 1, and $\frac{\delta f_m^v}{\delta t} = -0.157529215757$ for the other 4 crystals, resulting in a misfit of $6.5244 \cdot 10^{-4}$ % relative to the analytical solution.

We provide the D-REX_S_test program for reproducing this simple test in the Supplementary Material.

Table A1 – Available slip systems of abundant anisotropic mantle phases present in D-REX_S and D-REX_M.

| Slip system [uvw](hkl) | Olivine | Enstatite | Wadsleyite | Bridmanite | Post-Perovskite |
|---|---|---|---|---|---|
| [100](010) | ⋆ | | ⋆ | ⋆ | ⋆ |
| [100](001) | ⋆ | | ⋆ | ⋆ | ⋆ |
| [100](011) | | | ⋆ | | |
| [100](021) | | | ⋆ | | |
| [010](100) | | | | ⋆ | ⋆ |
| [010](001) | | | | ⋆ | ⋆ |
| [001](100) | ⋆ | ⋆ | | ⋆ | ⋆ |
| [001](010) | ⋆ | | ⋆ | ⋆ | ⋆ |
| [001]{$\bar{1}$10} | | | | ⋆ | ⋆ |

| | | | | | |
|---|---|---|---|---|---|
| $< \bar{1}10 > (001)$ | | | | ⋆ | 735 |
| $< 110 > \{\bar{1}10\}$ | | | | ⋆ | ⋆   736 |
| $\frac{1}{2} < 111 > \{101\}$ | | | ⋆ | | 737 |

**Appendix B. Aggregate elastic stiffness tensor**

At any P-T condition, the single crystal elastic moduli $C_{ij}^m$ are (in Voigt notation):

$$C_{ij}^m(P,T) = C_{ij}^m(P_0, T_0) + \partial C_{ij}^m/\partial P \cdot \Delta P + 0.5 \cdot \partial^2 C_{ij}^m/\partial P^2 \cdot \Delta P^2 + \partial C_{ij}^m/\partial T \cdot \Delta T + \partial^2 C_{ij}^m/\partial P\partial T \cdot \Delta P\Delta T \qquad (B1)$$

where the partial derivatives are listed in Table A2 and $\Delta P$ and $\Delta T$ are deviations from the room conditions ( $P_0 = 10^{-4}\ GPa, T_0 = 298\ K$; Mainprice, 2007). The compliance tensor is the inverse of the stiffness tensor, i.e., $\boldsymbol{S^m} = \boldsymbol{C^{m-1}}$. These tensors can be rotated as:

$$C_{ijkl}^v = a_{ip}a_{jq}a_{kr}a_{ls}C_{pqrs}^m \qquad (B2a)$$

$$S_{ijkl}^v = a_{ip}a_{jq}a_{kr}a_{ls}S_{pqrs}^m \qquad (B2b)$$

where $\boldsymbol{a}$ is the CDM defining the crystal orientation with respect to the external reference frame. The stiffness and compliance tensors of the crystal aggregate (i.e., rock) are then obtained by arithmetic averaging:

$$\boldsymbol{C^r} = \sum_{m=1}^{M} X_m \sum_{v=1}^{N} X_v\, \boldsymbol{C^v} \qquad (B3a)$$

$$\boldsymbol{S^r} = \sum_{m=1}^{M} X_m \sum_{v=1}^{N} X_v\, \boldsymbol{S^v} \qquad (B3b)$$

65 where $M = 2$ is the number of minerals in the two-phase mantle aggregates modelled in D-REX_M and D-REX_S, $X_m$ their

66 relative abundance, $N$ the number of crystals of each phase, $X_v$ the normalized crystal volume fraction ($\sum_{v=1}^{N} X_v = 1$). $\boldsymbol{C^r}$

67 and $(\boldsymbol{S^r})^{-1}$ correspond to the Voigt and Reuss average stiffness tensor of the rock, i.e., $\boldsymbol{C^r_{Voigt}}$ and $\boldsymbol{C^r_{Reuss}}$. More in general,

68 the average stiffness tensor resulting from the linear combination of the two end-members is found as:

69

70
$$\overline{\boldsymbol{C}}^r = X_{Voigt}\boldsymbol{C^r_{Voigt}} + \left(1 - X_{Voigt}\right)\boldsymbol{C^r_{Reuss}} \tag{B4}$$

71

72 where $X_{Voigt}$ is a free parameter defining the fraction of $\boldsymbol{C^r_{Voigt}}$ relative to $\boldsymbol{C^r_{Reuss}}$. It is clear that the Hill average stiffness

73 tensor $\boldsymbol{C^r_{Hill}}$ is obtained when $X_{Voigt} = 0.5$.

74 The isotropic component of the average stiffness tensor $\overline{\boldsymbol{C}}^r_{ISO}$ can be assembled as $\bar{C}^r_{ISO\_11} = \bar{C}^r_{ISO\_22} = \bar{C}^r_{ISO\_33} = K +$

75 $\frac{4}{3}G, \bar{C}^r_{ISO\_12} = \bar{C}^r_{ISO\_13} = \bar{C}^r_{ISO\_23} = K - \frac{2}{3}G, \bar{C}^r_{ISO\_44} = \bar{C}^r_{ISO\_55} = \bar{C}^r_{ISO\_66} = G$, knowing that:

$$K = \frac{[(\bar{C}^r_{11} + \bar{C}^r_{22} + \bar{C}^r_{33}) + 2(\bar{C}^r_{12} + \bar{C}^r_{13} + \bar{C}^r_{23})]}{9} \tag{B5a}$$

$$G = \frac{[(\bar{C}^r_{11} + \bar{C}^r_{22} + \bar{C}^r_{33}) - (\bar{C}^r_{12} + \bar{C}^r_{13} + \bar{C}^r_{23}) + 3(\bar{C}^r_{44} + \bar{C}^r_{55} + \bar{C}^r_{66})]}{15} \tag{B5b}$$

81 The isotropic component of the aggregate stiffness tensor $\overline{\boldsymbol{C}}^r_{ISO}$ is then replaced with the isotropic stiffness tensor $\boldsymbol{C}^{MMA\ EoS}_{ISO}$

82 obtained using the bulk and shear moduli from the lookup tables computed with the MMA_EoS software (Chust et al., 2017)

83 as:

$$\overline{\boldsymbol{C}}^r_{MMA\_EoS} = \overline{\boldsymbol{C}}^r + \boldsymbol{C}^{MMA\_EoS}_{ISO} - \overline{\boldsymbol{C}}^r_{ISO} \tag{B6}$$

89 Table B1: Composition (in mol%) used in MMA_EoS to compute lookup tables for different mantle and basalt composition.

| Lithology | Index | $SiO_2$ | $MgO$ | $FeO$ | $CaO$ | $Al_2O_3$ | $Na_2O$ | Source[†] |
|-----------|-------|---------|-------|-------|-------|-----------|---------|-----------|
| Dunite | 1 | 40.43 | 42.45 | 16.07 | 0.51 | 0.51 | 0.04 | (1) |
| Harzburgite | 2 | 36.54 | 56.17 | 5.71 | 0.99 | 0.59 | 0.00 | (2) |
| Pyrolite | 3 | 38.71 | 49.85 | 6.17 | 2.94 | 2.22 | 0.11 | (3) |

| | | | | | | | | | | | |
|---|---|---|---|---|---|---|---|---|---|---|---|
| Basalt | 4 | 51.76 | 15.11 | 6.59 | 14.39 | 10.39 | 1.76 | | (2) | | |
| Pyroxenite | 5 | 46.46 | 16.99 | 7.95 | 11.70 | 15.48 | 1.43 | | (4) | | |

[†] (1) Kelemen and Ghiorso, 1986; (2) Chemia et al., 2015; (3) Workman and Hart, 2005; (4) Hirschmann et al., 2003.

Table B2 - Elastic moduli (GPa) and their P–T derivatives of the mineral phases present in ECOMAN. Temperature derivatives are scaled by $10^2$ GPa K$^{-1}$, and cross derivatives by $10^{-3}$ K$^{-1}$.

| Mineral | Index | Moduli | C11 | C22 | C33 | C12 | C13 | C23 | C44 | C55 | C66 | Source[†] |
|---|---|---|---|---|---|---|---|---|---|---|---|---|
| Olivine | 1 | $M$ | 320.71 | 197.25 | 234.32 | 69.84 | 71.22 | 74.80 | 63.77 | 77.67 | 78.36 | (1) |
| | | $\partial M/\partial P$ | 5.41 | 5.26 | 3.78 | 1.88 | 1.53 | 1.60 | 1.08 | 1.42 | 1.80 | (2) |
| | | $\partial M/\partial T$ | -4.02 | -3.10 | -3.53 | -1.14 | -0.96 | -0.72 | -1.26 | -1.30 | -1.56 | (1) |
| Enstatite | 2 | $M$ | 236.90 | 180.50 | 230.40 | 79.60 | 63.20 | 56.80 | 84.30 | 79.40 | 80.10 | (3) |
| | | $\partial M/\partial P$ | 10.27 | 8.87 | 11.7 | 6.22 | 6.63 | 7.26 | 1.23 | 0.75 | 2.78 | (3) |
| | | $\partial^2 M/\partial P^2$ | -0.47 | -0.38 | -0.53 | -0.33 | -0.26 | -0.31 | 0 | 0 | 0 | (3) |
| | | $\partial M/\partial T$ | -3.64 | -3.43 | -5.70 | -1.52 | -2.29 | -1.63 | -1.23 | -1.58 | -1.51 | (4) |
| Wadsleyite | 3 | $M$ | 370.50 | 367.70 | 272.40 | 65.60 | 95.20 | 105.10 | 111.20 | 122.50 | 103.10 | (5) |
| | | $\partial M/\partial P$ | 5.21 | 6.83 | 8.06 | 4.06 | 3.30 | 3.30 | 1.39 | 0.00 | 1.88 | (5) |
| | | $\partial M/\partial T$ | -4.02 | -3.10 | -3.53 | -1.14 | -0.96 | -0.72 | -1.26 | -1.30 | -1.56 | (1) |
| Wadsleyite Fe = 0.94 H2O = 0.15 wt% | 4 | $M$ | 341.00 | 358.00 | 224.00 | 75.00 | 99.00 | 102.00 | 106.00 | 109.00 | 90.80 | (6) |
| | | $\partial M/\partial P$ | 7.80 | 5.37 | 10.5 | 4.60 | 2.33 | 4.20 | 0.87 | 0.84 | 2.43 | (6) |
| | | $\partial^2 M/\partial P^2$ | -0.26 | 0 | -0.46 | -0.11 | 0 | -0.14 | 0 | 0 | -0.053 | (6) |
| | | $\partial M/\partial T$ | -2.90 | -3.60 | -4.00 | -0.60 | -1.90 | -1.20 | -1.80 | -1.10 | -1.30 | (6) |
| Ringwoodite | 5 | $M$ | 329.00 | - | - | 118.00 | - | - | 130.00 | - | - | (7) |
| | | $\partial M/\partial P$ | 6.20 | - | - | 0.80 | - | - | 2.80 | - | - | (7) |
| | | $\partial M/\partial T$ | -4.90 | - | - | -0.70 | - | - | -1.30 | - | - | (7) |
| Garnet | 6 | $M$ | 299.10 | - | - | 106.70 | - | - | 93.70 | - | - | (8) |
| | | $\partial M/\partial P$ | 6.54 | - | - | 2.87 | - | - | 1.72 | - | - | (8) |
| | | $\partial M/\partial T$ | -3.05 | - | - | -0.58 | - | - | -0.71 | - | - | (9) |
| Bridgmanite | 7 | $M$ | 484.00 | 542.00 | 477.00 | 146.00 | 146.00 | 162.00 | 195.00 | 172.00 | 151.00 | (9) |
| | | $\partial M/\partial P$ | 3.40 | 5.30 | 5.20 | 3.30 | 2.40 | 2.50 | 1.40 | 0.80 | 1.40 | (10) |
| | | $\partial M/\partial T$ | -1.80 | -3.30 | -2.20 | -0.40 | -0.20 | -0.50 | -2.20 | -0.60 | -1.60 | (10) |
| | | $\partial^2 M/\partial P \partial T$ | -0.017 | -0.016 | -0.016 | -0.006 | -0.006 | -0.007 | -0.0005 | -0.0005 | -0.004 | (10) |
| Bridgmanite P0 = 34 GPa T0 = 1500 K | 8 | $M$ | 581.42 | 669.06 | 626.31 | 249.02 | 212.93 | 239.01 | 228.63 | 220.26 | 183.89 | (11) |
| | | $\partial M/\partial P$ | 3.73 | 5.54 | 5.47 | 3.30 | 2.48 | 2.53 | 1.38 | 0.85 | 1.56 | (11) |
| | | $\partial M/\partial T$ | -3.99 | -5.53 | -4.32 | -1.76 | -0.64 | -0.26 | -2.12 | -1.25 | -1.96 | (11) |
| | | $\partial^2 M/\partial P \partial T$ | -0.16 | -0.044 | -0.051 | -0.017 | -0.06 | -0.035 | -0.017 | -0.044 | -0.042 | (10) |
| Bridgmanite P0 = 48 GPa | 9 | $M$ | 601.83 | 698.17 | 652.27 | 271.81 | 236.00 | 265.58 | 233.53 | 205.41 | 190.43 | (11) |
| | | $\partial M/\partial P$ | 3.73 | 5.54 | 5.47 | 3.30 | 2.48 | 2.53 | 1.38 | 0.85 | 1.56 | (11) |

| | | | | | | | | | | | | | |
|---|---|---|---|---|---|---|---|---|---|---|---|---|---|
| T0 = 1500 K | | $\partial M/\partial T$ | -3.14 | -5.94 | -4.53 | -1.94 | -0.55 | -0.59 | -2.15 | -1.26 | -2.50 | (11) |
| | | $\partial^2 M/\partial P\partial T$ | -0.66 | -0.026 | -0.034 | -0.158 | -0.008 | -0.160 | -0.001 | -0.036 | -0.002 | (11) |
| Ferropericlase | 10 | $M$ | 300.0 | - | - | 93.60 | - | - | 147.0 | - | - | (12) |
| | | $\partial M/\partial P$ | 9.56 | - | - | 1.45 | - | - | 1.03 | - | - | (12) |
| | | $\partial M/\partial T$ | -5.98 | - | - | 0.89 | - | - | -0.88 | - | - | (12) |
| | | $\partial^2 M/\partial P\partial T$ | 0.56 | - | - | 0.06 | - | - | 0.20 | - | - | (12) |
| post-Perovskite | 11 | $M$ | 1124.75 | 884.22 | 1124.32 | 350.69 | 287.53 | 461.05 | 254.57 | 231.19 | 344.05 | (13) |
| P0 = 99.95 GPa | | $\partial M/\partial P$ | 5.85 | 2.70 | 5.44 | 3.06 | 3.00 | 2.33 | 1.37 | 1.27 | 2.06 | (13) |
| T0 = 1500 K | | $\partial M/\partial T$ | 1.01 | -5.24 | -4.90 | -0.92 | -0.51 | -1.78 | -0.81 | -2.33 | -2.19 | (13) |
| | | $\partial^2 M/\partial P\partial T$ | -0.937 | 0.709 | 0.797 | 0.60 | -0.801 | 0.692 | -0.107 | -0.186 | -0.16 | (13) |

[†] (1) Isaak, 1992; (2) Zha et al., 1998; (3) Chai et al., 1992; (4) Jackson et al., 2007; (5) Zha et al., 1997; (6) Zhou et al., 2022; (7) Sinogeikin et al., 2003; (8) Chai and Brown, 1997; (9) Sinogeikin and Bass, 2002; (10) Wentzcovitch et al., 2004. Derivatives at (100 GPa, 300 K); (11) Zhang et al., 2013. Derivatives extrapolated by interpolation; (12) Karki et al., 2000; (13) Zhang et al., 2016. Derivatives extrapolated by interpolation.

## Appendix C. Velocity gradient tensor in polar coordinates

In polar grids the velocity field is typically defined by the longitudinal, radial and, in 3D, colatitudinal components $\vec{V}_{(\phi,r,\theta)} = (V_\phi, V_r, V_\theta)$. However, the crystals' orientation and their rotation are defined relative to the external Cartesian reference frame. As such, the velocity gradient tensor must be computed in Cartesian coordinate system. This is done by rotating the velocity field from spherical to Cartesian coordinates as:

$$\begin{bmatrix} V_x \\ V_y \\ V_z \end{bmatrix} = \begin{bmatrix} -\sin\phi & \sin\theta\cos\phi & \cos\theta\cos\phi \\ \cos\phi & \sin\theta\sin\phi & \cos\theta\sin\phi \\ 0 & \cos\theta & -\sin\theta \end{bmatrix} \begin{bmatrix} V_\phi \\ V_r \\ V_\theta \end{bmatrix} \qquad (C1)$$

Using the finite difference approach, the spatial gradients of the Cartesian velocity components are computed on the spherical grid as:

$$\boldsymbol{L}_{\frac{(x,y,z)}{(\phi,r,\theta)}} = \frac{\partial \vec{\boldsymbol{V}}_{(x,y,z)}}{\partial(\phi,r,\theta)} = \begin{bmatrix} \dfrac{\partial V_x}{\partial \phi} & \dfrac{\partial V_x}{\partial r} & \dfrac{\partial V_x}{\partial \theta} \\ \dfrac{\partial V_y}{\partial \phi} & \dfrac{\partial V_y}{\partial r} & \dfrac{\partial V_y}{\partial \theta} \\ \dfrac{\partial V_z}{\partial \phi} & \dfrac{\partial V_z}{\partial r} & \dfrac{\partial V_z}{\partial \theta} \end{bmatrix} \qquad (C2)$$

Defining $\boldsymbol{J}$ as the Jacobian matrix:

$$J = \frac{\partial(\phi, r, \theta)}{\partial(x, y, z)} = \begin{bmatrix} \frac{\partial\phi}{\partial x} & \frac{\partial\phi}{\partial y} & \frac{\partial\phi}{\partial z} \\ \frac{\partial r}{\partial x} & \frac{\partial r}{\partial y} & \frac{\partial r}{\partial z} \\ \frac{\partial\theta}{\partial x} & \frac{\partial\theta}{\partial y} & \frac{\partial\theta}{\partial z} \end{bmatrix} = \begin{bmatrix} -sin\phi/(r \cdot sin\theta) & cos\phi/(r \cdot sin\theta) & 0 \\ sin\theta cos\phi & sin\theta sin\phi & cos\theta \\ cos\theta cos\phi/r & cos\theta si & \phi/r & -sin\theta/r \end{bmatrix} \qquad (C3)$$

24    the velocity gradient tensor in Cartesian coordinates is $\mathbf{L}_{\underset{(x,y,z)}{(x,y,z)}} = \mathbf{L}_{\underset{(\phi,r,\theta)}{(x,y,z)}} \mathbf{J}$:

$$\mathbf{L}_{\underset{(x,y,z)}{(x,y,z)}} = \frac{\partial\vec{V}_{(x,y,z)}}{\partial(x, y, z)} = \begin{bmatrix} \frac{\partial V_x}{\partial x} & \frac{\partial V_x}{\partial y} & \frac{\partial V_x}{\partial z} \\ \frac{\partial V_y}{\partial x} & \frac{\partial V_y}{\partial y} & \frac{\partial V_y}{\partial z} \\ \frac{\partial V_z}{\partial x} & \frac{\partial V_z}{\partial y} & \frac{\partial V_z}{\partial z} \end{bmatrix} = \begin{bmatrix} \frac{\partial V_x}{\partial \phi} & \frac{\partial V_x}{\partial r} & \frac{\partial V_x}{\partial \theta} \\ \frac{\partial V_y}{\partial \phi} & \frac{\partial V_y}{\partial r} & \frac{\partial V_y}{\partial \theta} \\ \frac{\partial V_z}{\partial \phi} & \frac{\partial V_z}{\partial r} & \frac{\partial V_z}{\partial \theta} \end{bmatrix} \begin{bmatrix} \frac{\partial\phi}{\partial x} & \frac{\partial\phi}{\partial y} & \frac{\partial\phi}{\partial z} \\ \frac{\partial r}{\partial x} & \frac{\partial r}{\partial y} & \frac{\partial r}{\partial z} \\ \frac{\partial\theta}{\partial x} & \frac{\partial\theta}{\partial y} & \frac{\partial\theta}{\partial z} \end{bmatrix} \qquad (C4)$$

**Appendix D. Aggregate rotation due to fluid body rotation**

When multiple creep mechanisms are active, we assume that only the fraction of deformation accommodated by dislocation

creep contributes to the LPO development. This fraction of the total deformation is defined as $F_d = \eta_{eff}/\eta_{disl}$, where

$0 \leq F_d \leq 1$ is interpolated from the large-scale geodynamic model, $\eta_{eff}$ is the effective viscosity resulting from the harmonic

average of the viscosities from the flow law of different creep mechanisms, $\eta_{disl}$ the viscosity from the dislocation creep flow

law. At a given timestep $\Delta t$, crystal rotation according to the D-REX model is applied for the fraction $\Delta t_{disl} = \Delta t \cdot F_d$, while

for the remaining time $\Delta t_{diff} = \Delta t \cdot (1 - F_d)$ we follow the numerical study of Hedjazian et al. (2017) and apply fluid body

rotation by multiplying the CDM of the aggregates with the rotation of the FSE. In this way, we preserve and do not alter the

strength of the LPO.

Given the orthogonal CDMs defining the original and final crystal orientation $\mathbf{a}^t$ and $\mathbf{a}^{t+\Delta t_{diff}}$, and those defining the original

and final FSE semiaxes orientation $\mathbf{v}^t$ and $\mathbf{v}^{t+\Delta t_{diff}}$ (i.e., the eigenvectors of, respectively, the left stretch tensor $\mathbf{LS}^t =$

$\mathbf{F}^t(\mathbf{F}^t)^T$ and $\mathbf{LS}^{t+\Delta t_{diff}} = \mathbf{F}^{t+\Delta t_{diff}}(\mathbf{F}^{t+\Delta t_{diff}})^T$), the following relation implies that after fluid body rotation the orientation

of the crystal relative to FSE semiaxes must remain constant:

$$\mathbf{a}^{t+\Delta t_{diff}} \mathbf{v}^{t+\Delta t_{diff}} = \mathbf{a}^t \mathbf{v}^t \qquad (D1)$$

The new crystal orientation is then:

$$\mathbf{a}^{t+\Delta t_{diff}} = \mathbf{a}^t \mathbf{v}^t (\mathbf{v}^{t+\Delta t_{diff}})^{-1} = \mathbf{a}^t \mathbf{v}^t (\mathbf{v}^{t+\Delta t_{diff}})^T \qquad (D2)$$

**Additional References**

M. Chai and J. Brown. The elastic constants of a pyrope-grossular-almandine garnet to 20 GPa. Geophys. Res. Lett., 24(5):523–526, 1997.

M. Chai, J. Brown, and L. Slutski. The elastic constants of an aluminous orthopyroxene to 12.5 GPa. J. Geophys. Res., 102(14):779–785, 1992.

Z. Chemia, D. Dolejš, and G. Steinle-Neumann. Thermal effects of variable material properties and metamorphic reactions in a three-component subducting slab. J. Geo- phys. Res., 120:6823–6845, 2015.

M. Hirschmann, T. Kogiso, M. Baker, and E. Stopler. Alkalic magmas generated by partial melting of garnet pyroxenite. Geology, 31:481–484, 2003.

D. Isaak. High-temperature elasticity of iron-bearing olivines. J. Geophys. Res., 97: 1871–1885, 1992.

J. Jackson, S. Sinogeikin, and J. Bass. Sound velocities and single-crystal elasticity of orthoenstatite to 1073 K at ambient pressure. Phys. Earth Planet. Int., 171:1–12, 2007.

B. Karki, R. Wentzcovitch, S. de Gironcoli, and S. Baroni. High-pressure lattice dynamics and thermoelasticity of MgO. Phys. Rev. B, 61:8793–8800, 2000.

P. Kelemen and M. Ghiorso. Assimilation of peridotite in zoned calc-alkaline plutonic complexes: evidence from the big jim complex, Washington Cascades. Contrib. Min., 94:12–28, 1986.

David Mainprice. Seismic Anisotropy of the Deep Earth from a Mineral and Rock Physics Perspective. Treatise on Geophysics, 2, Editor: G. Schubert, Elsevier, pp.437-491, 2007.

S. Sinogeikin and J. Bass. Elasticity of pyrope and majorite – pyrope solid solutions to high temperatures. Earth Planet. Sci. Lett., 203:549–555, 2002.

S. Sinogeikin, J. Bass, and K. Katsura. Single-crystal elasticity of ringwoodite to high pressures and high temperatures: implications for 520 km seismic discontinuity. Phys. Earth Planet. Int., 136:41–66, 2003.

R. Wentzcovitch, B. Karki, M. Cococcioni, and S. de Gironcoli. Properties of MgSiO3-perovskite: Insights on the nature of the earth's lower mantle. Phys. Rev. Lett., 92, 2004.

R. Workman and S. Hart. Major and trace element composition of the depleted morb mantle. Earth Planet. Sci. Lett., 231:53–72, 2005.

C. Zha, T. Duffy, H. Mao, R. Downs, R. Hemley, and D. Weidner. Single-crystal elasticity of β-Mg2SiO4, to the pressure of the 410 km seismic discontinuity in the earth's mantle. Earth Planet. Sci. Lett., 147:E9–E15, 1997.

C. Zha, T. Duffy, R. Downs, H. Mao, and R. Hemley. Brillouin scattering and x-ray diffraction of San Carlos olivine: direct pressure determination to 32 GPa. Earth Planet. Sci. Lett., 159:25–33, 1998.

Z. Zhang, L. Stixrude, and J. Brodholt. Elastic properties of MgSiO3-perovskite under lower mantle conditions and the composition of the deep earth. Earth Planet. Sci. Lett., 379:1–12, 2013.

81  Z. Zhang, S. Cottaar, L. Tao, S. Stackhouse, and B. Militzer. High-pressure, temperature elasticity of Fe- and Al-bearing

82      MgSiO3: Implications for the earth's lower mantle. Earth Planet. Sci. Lett., 434:264–273, 2016.

83  W.-Y. Zhou, M. Hao, J. Zhang, B. Chen, R. Wang, and B. Schamndt. Constraining composition and temperature variations in

84      the mantle transition zone. Nat. Comm., 13:1094, doi: 10.1038/s41467-022-28709-7, 2022.