# Peer review of "ECOMAN: an open-source package for geodynamic and"

_EGUsphere, 2024_

## Author Comment (AC1)

**General comments**

The manuscript describes a suite of tools for the modelling of mechanical anisotropy, both its development for a given mineralogical model under geodynamic strain and the evaluation of various seismic parameters. The tools are (mostly) not new, but have been collated and refactored to make building them into a workflow simpler. The process of this workflow has also been nicely documented in an extensive manual for the suite, and several instructive examples.

Such activities are often not well documented and shared, so I appreciate the extra efforts the authors have made to make their work accessible and useful to the community. While there are of course some limitations to all of the tools in the suite, there are a large range of important, interesting problems that can be tackled with these tools either singly or together which make the publication of a manuscript describing them worthy.

I have some specific comments which I think that the authors should address before considering the manuscript finalised. These are (mostly) focussed on the seismic end of the problem. However, I would also like to endorse the general comments made by RC1, and agree that the two missing sections identified would make the manuscript much more complete. A version of record on Zenodo (or similar) would also be sensible.

*Thanks for the comments, we have carefully addressed them as reported in the replies to RC1's comments, and as the following:*

**Specific comments**

Line 30-33, sentence beginning "Seismological and ...": This implies that the suite presented does something different, which – as far as I can see – is not really the case. The calculations of (for example) SKS-splitting here is not done on the original geodynamics grid, for example. I'd also argue that this is not really as definitively a problem as stated, either.

*The true and most important novelty of ECOMAN is that it allows to perform P- and/or S-wave isotropic and anisotropic tomographies using real datasets or synthetic datasets generated from the geodynamically-derived elastic tensors and densities. To our knowledge, this is unprecedented, except for the isotropic approach employed by some groups and that is cited in the following sentences (Styles et al., 2011; Schubert et al., 2012; Maguire et al., 2018).*
*We respect the opinion of the reviewer, but we believe that the problems mentioned in that sentence (in particular, the validation of geodynamic models against observations, and the interpretation of the imaged seismic structures in terms of geodynamic processes) are of paramount importance, and indeed dedicated sessions are routinely organized in the main scientific meetings (e.g., https://meetingorganizer.copernicus.org/EGU24/session/49108).*

Line 81, s/b "At the same time ...": These are not the earliest examples of 3D anisotropy inversions. There are a range of these in the literature, dating back (at least) to Panning et al (GJI, 2006) for global models, shear-wave splitting (Abt and Fischer, 2008; Long et al, 2008; Wookey, 2012), and surface waves (Debayle et al, 2005), to name but a few.

*Here we meant that the methods recently developed by a few groups (us, Prof. Zhao's and Prof. Plomerova's) are capable of simultaneously recovering for the isotropic velocity anomalies and seismic anisotropy for arbitrarily oriented fabrics. Previous methodologies such as those employed by Panning et al., (GJI, 2006) and Debayle et al., (2005) are capable of inverting for either the radial or azimuthal components of the shear wave anisotropy, while those presented by Abt and Fisher (2008), Long et al. (2008), and Mondal and Long (2019, GJI) can only retrieve the anisotropic component of S-waves.*
*Nevertheless, we acknowledge the importance of these previous studies and mention them in the text.*

Line 279. s/b "the fraction of ...": It might be worth noting that the tensor 'fractions' from this approach have been suggested to not be a good approximation (Tape and Tape, Journal of Elasticity, 2024).

*We are aware of the work by Tape and Tape, 2024 (see their acknowledgements), and we clarify this problem in section 3.2 about the current software limitations.*

Section 2.3.1: What equations are solved to calculated the splitting? How is the frequency included?

*The methodology behind the SKS splitting calculation is now briefly explained in section 2.3.1. The methodology details have been already described in Schulte-Pelkum and Blackmann, 2003 by the software developer.*

Line 445, s/b: The underdetermination is inherent in the problem, at best MC sampling might be able to constrain the uncertainty due to the underdetermination, which is I suspect what the authors mean by this, but it should be clearer.

*Right. We have clarified this aspect accordingly.*

**Technical corrections**

Lines 318, 438: font issues

*This was a problem that occurred during PDF conversion, now fixed.*

Line 39: 'in' -> 'of'

*Corrected*

Line 415: It is more useful to describe MATLAB and Python as *interpreted* languages, rather than high level. C is also technically a high level language, and is as fast or faster than Fortran.

*Corrected*

Line 417: '100.000s' -> '100,000s', or use words to avoid decimal point confusion.

*Corrected*

Line 424: 'it's' -> 'its'

*Corrected*

Line 451: 'surface waves' -> 'surface wave'

*Corrected*

*Thanks for the valuable feedback*

*Manuele Faccenda, on behalf of all co-authors*

---

## Author Comment (AC2)

**General Comments**

This paper presents a suit of tools related to the forward and inverse modelling of mechanical anisotropy. The capabilities which these tools provide are in my opinion a very important contribution to scientific progress within the scope of this journal. The paper described the purpose of the individual tools and explains at a high level what kind of changes level changes which where made (in case of being build on previous work), some considerations on how to use it and shows example explanations.

Unfortunately, there are, in my opinion, two major sections missing in this paper. The first sections is a proper methods section, which describes for each tool how it works. For example, what are the equations which are being solved. If it is based on previous work (such as D-REX), what changes have been made to the methods (more detailed then the current description) and how does this align with the assumptions made in the original code. What values of different parameters are used for computing for example wadsleyite or bridgmanite (both in D-REX and in the computation of the elastic tensor).

*D-REX_M and D-REX_S are built upon the original D-REX code, which has been extensively described by Kaminski and Ribe, 2001, and Kaminski et al., 2004. Thus, we think it is not necessary to report them here. The existing strategy for computing the LPO evolution of Olivine and Pyroxene crystals has been similarly applied to Wadsleyite, Bridgmanite and post-Perovskite using the corresponding slip systems as listed in Table A1. For each of these new phases it is possible to define in the D-REX_M and D-REX_S input file the same free parameters of the D-REX model ($M^*, \lambda^*, \chi^*$, together with in addition the power-law exponent) as for upper mantle aggregates. This is now better clarified in section 2.1.*

*Nevertheless, we agree with the reviewer that the further developments included in the D-REX model should be better explained, and accordingly in the new version of the article we now include:*

- *Appendix A, where we test the numerical solution of the employed D-REX model against the analytical solution derived by Fraters and Billen (2021), and report the available slip systems for the different anisotropic phases in Table A1.*
- *Appendix B, where we describe how the elastic tensors are computed as a function of the local P-T conditions, crystal orientation and volume fraction, modal abundance and aggregate composition. In this section we also include Table B1 with the composition of the 5 mafic and ultramafic lithologies for which lookup tables of Vp, Vs and density have been computed with MMa_EoS, and Table B2 with available the single crystal elastic moduli and their P-T derivatives.*
- *Appendix C, where we explain how the velocity gradient is computed in Cartesian coordinate system when the grid is in polar coordinates*
- *Appendix D, where it is explained how the fluid body rotation is applied when the deformation is accommodated by mechanisms other than dislocation creep.*
- *new Figure 2, where we show the P-T distribution of crystal aggregates in a pyrolytic mantle*

- *new Figure 4, where we show sensitivity tests of olivine fabrics and demonstrated that with a suitable choice of the D-REX parameters it is possible to produce either weak or strong olivine fabrics as observed in natural and experimental samples.*

A second section which is missing is a benchmark section, to show that what is stated in the methods section actually works as intended. These should be at least a small suite of simple tests. For example computing the CPO in a simple shear environment, which can be matched against analytic results, the results of other codes and/or experiments. The only benchmarks which are shown are performance benchmarks.

*As explained above, we now include results from a simple benchmark test in Appendix A and several sensitivity tests of the olivine fabric evolution in simple shear reported in Figure 4. It is important to note that the cookbooks included in ECOMAN and partly shown in Figs. 6-9 were effectively designed to function as benchmarks for complex tests. This is why we have not shown simpler benchmarks, but indeed this comment has proven to be quite constructive as it gave us the possibility to demonstrate the ability of the D-REX model to reproduce high-strain olivine fabrics.*

Because these two sections are missing it is not possible to properly review the paper, since the authors do not show what they have done exactly and that they have done it correctly. Although the code is open-source, and it could in theory be checked by looking into the code and creating and running benchmarks yourself, I do think this should be part of the paper.

*We hope that the added material and clarifications will help in better evaluating the article.*

**Specific comments**

- The geodynamics code which are used for the examples should be mentioned. Looking at the references, it seems like for some of them I3MG is used, but this is not explicitly mentioned and it is not clear for all of them.

*The flow fields in the 2D examples are from analytical solutions, and MATLAB scripts are included in ECOMAN. The code used for the 3D examples in Cartesian coordinates is I3MG, which has been modified to account for spherical grids as described in Faccenda and VanderBeek, 2023. This is now clarified in the captions of Figs. 6-10.*

- Github is not a software archive, since it can easily be removed or changed. It is good to mention it, *but you will also need a doi of for the software. You can get this for example from Zenodo.*

*Thanks for the suggestion, we have created DOIs which are now indicated in the code availability section.*

- The repository has a license (MIT), a changlog document, a proper versioning scheme and a comprehensive user manual. I assume that this paper is about version 2.0, but this should be explicitly stated in the paper. I could not find any tests or benchmarks in the repository, although it could be that the cookbooks function as tests.

*In the code availability section, it is now stated that the version of ECOMAN described in this article is 2.0. As stated above, the cookbooks function as tests and benchmarks.*

**Technical corrections**

line 135-137: Name and cite studies which are used together with the lookup tables. Also, it just states that the fabric selection is P-T dependent, so no water dependence? To come back to the general point, it needs to be explained how these fabrics are selected. A graph or table would be nice, if feasible.

*In the Appendix we now include new Tables A1, B1 and B2 about the available slip systems, the composition used in MMA-EoS for the lookup tables, and single crystal elastic tensors and their P-T derivatives, including all the source studies. The depth/density crossovers defining major phase transitions and the parametrized phase boundary for Pv – pPv are indicated at the beginning of section 2.1.*

*At present we have not included an Olivine fabric phase diagram as in Fraters and Billen 2021. This is planned in future release of the software, as we state in section 3.2.*

line 361-362: Workstation is a vague term, mention CPU and ram needed to do that that calculation in the stated time.

*These details are now indicated.*

line 428: I assume that the authors mean spreading memory load over several nodes, not CPU's.

*Corrected*

*Thanks for the valuable feedback*

*Manuele Faccenda, on behalf of all co-authors*

---

## Author Response (AR2)

*Dear Editor and Reviewer #1,*

*thanks for your comments.*
*Please find below our replies in italic and highlighted in yellow.*

*Best regards,*

*Manuele Faccenda*

Dear Manuele Faccenda and co-authors,

We have now heard back from Referee #1, who had raised some concerns during the first round of reviews, but found the revision significantly improved. Before accepting the manuscript, I would like to give you the opportunity to address the technical comments from Referee #1. In addition, the similarity report we run as a routine check detected that there is some overlap in text between the ECOMAN manual and this manuscript (about 10% overall, mostly in Sections 2.1.2 and 3.1), and it would be great if you could rephrase these parts. (Finally, I also noticed a typo while reading: The reference Banghert et al., 2020 should be Bangerth et al., 2020.) Thank you for being so patient, and for submitting your work to Solid Earth!

Sincerely,
Juliane Dannberg

*Dear Editor,*
*the content in sections 2.1.2 and 3.1 have been removed from the manual, which in fact was redundant. The typo has been fixed.*
*I have also added the additional reference Rappisi et al., 2024, GJI, an article from my group and collaborators that was recently accepted and that shows an application of D-REX_M to a geodynamic model in polar coordinates.*

**Replies to Reviewer #1**

The authors significantly improved the manuscript and reviewing it now. I am happy with the extra information and explanation the authors provided. I only have specific and technical comments left. Other than that I would be happy to see this paper published.

Specific/Technical comments

- Line 132: Just to make sure I understand what is happening: when an LPO particle moves from for example a location where it is in the Wadsleyite phase (2), to a location where it is in the ringwoodite phase (3), it is assumed that no LPO is preserved and the LPO is set to be random?

*When a particle enters the stability field of another anisotropic phase, the LPO can be either reset or retained (which implies axisymmetric topotactical growth).*
*This is now clarified at lines 195-196, and better explained in the manual (see section 4.6).*

- As a general note, the scaling tests only go up to 8 cpu cores on a single workstation. This scales well, and it good to show, but it only really shows the shared memory scaling (OpenMP

I assume), but in line 276-277 HPC clusters are mentioned. These benchmarks do not show scaling on HPC clusters, which usually use MPI communication over many nodes. The authors do not show scaling for this purpose. Given that these tools seem to be mostly focused on post-processing, that may not be needed. So the authors should either clarify this in the text or add information to support their claim of HPC MPI scaling.

*The HPE Superdome Flex is a full shared memory machine which performs at best if there is no communication among the CPUs. Otherwise, the runtime increases notably due to communication among the CPU nodes. In other words, the superdome architecture allows to have the illusion that the memory is shared, but physically it is distributed on the 8 different sockets.*
*As D-REX_M is parallelized with both MPI and OpenMP, and to exploit the best performance of the Superdome, the scaling tests shown in Fig. 5 are performed distributing the load across up to 8 CPUs with 28 cores each. Thus, in these tests each CPU works independently from the others, and they effectively represent HPC MPI scaling tests, although with a limited number of nodes (but that's what we have currently available in our department). This is clarified in the caption of Fig. 5.*

- Figure 5 mentions CPU nodes, this is a bit confusing. I think this should be either be just CPU's or CPU cores.

*The 8 CPUs present in the Superdome are actually different nodes (sockets) with their own memory, which can be shared thanks to fast interconnections.*

- Figure 5: The purple line is very hard to see since it is behind the red line. It may be good to make the purple line a bit thicker to improve clarity

*The thickness of the purple lines has been doubled, which improves their visibility as suggested by the reviewer.*